# Lipschitz Continuity in Deep Learning: A Systematic Review of Theoretical Foundations, Estimation Methods, Regularization Approaches, and Certifiable Robustness

**Róisín Luo**\*
*Research Ireland – Centre for Research Training in AI (CRT-AI)*
*J.E. Cairnes School of Business & Economics*
*School of Computer Science*
*University of Galway, Ireland*

**James McDermott**
*Research Ireland – Centre for Research Training in AI (CRT-AI)*
*School of Computer Science*
*University of Galway, Ireland*

**Colm O'Riordan**
*Research Ireland – Centre for Research Training in AI (CRT-AI)*
*School of Computer Science*
*University of Galway, Ireland*

**Reviewed on OpenReview:** *https://openreview.net/forum?id=pRZORKl11f*
*Code: https://github.com/roisincrtai/lipschitz_survey*

## Abstract

Lipschitz continuity is a fundamental property of neural networks that characterizes their sensitivity to input perturbations. It plays a pivotal role in deep learning, governing **robustness**, **generalization** and **optimization dynamics**. Despite its importance, research on Lipschitz continuity is scattered across various domains, lacking a unified perspective. This paper addresses this gap by providing a systematic review of Lipschitz continuity in deep learning. We explore its **theoretical foundations**, **estimation methods**, **regularization approaches**, and **certifiable robustness**. By reviewing existing research through the lens of Lipschitz continuity, this survey serves as a comprehensive reference for researchers and practitioners seeking a deeper understanding of Lipschitz continuity and its implications in deep learning.

## 1 Introduction

Deep learning has achieved remarkable success across various domains, including computer vision, natural language processing, and graph-based learning. Advances in state-of-the-art architectures, such as convolutional neural networks (CNNs) (Krizhevsky et al., 2012), transformers (Vaswani et al., 2017; Brown et al., 2020), and graph neural networks (GNNs) (Kipf & Welling, 2017), as well as their large-scale applications, including large language models (LLMs) (Brown et al., 2020; Chowdhery et al., 2023; Touvron et al., 2023; OpenAI et al., 2024; DeepSeek-AI et al., 2025) and large vision-language models (LVLMs) (Radford et al., 2021; Li et al., 2022; Team et al., 2025), have significantly expanded the frontiers of deep learning, enabling powerful capabilities across a wide range of tasks.

Beyond their remarkable success, ensuring safety (Amodei et al., 2016), robustness (Madry et al., 2018) and generalization, remains a critical challenge. These properties are intrinsically linked to the sensitivity of neu-

---

\*Correspondence to: `roisincrtai@gmail.com`

ral networks to input perturbations (Madry et al., 2018; Tsipras et al., 2019; Hendrycks & Dietterich, 2019; Amerehi & Healy, 2025), which can significantly impact their reliability in real-world applications. For instance, adversarial vulnerabilities (Goodfellow et al., 2015; Carlini & Wagner, 2017), and out-of-distribution (OOD) generalization (Sokolić et al., 2017; Neyshabur et al., 2017; Bartlett et al., 2017; Hendrycks et al., 2021; Song et al., 2026) are major concerns in developing reliable deep learning models. A fundamental mathematical concept that characterizes network sensitivity and governs these properties is **Lipschitz continuity**, which measures the upper bound of the outputs of neural networks to inputs.

Although numerous studies have explored Lipschitz continuity from both theoretical (Bartlett et al., 2017; Luo et al., 2025a) and applied (Arjovsky et al., 2017; Gulrajani et al., 2017; Miyato et al., 2018) perspectives, a unified survey that consolidates these insights is still lacking. Existing works primarily focus on isolated aspects, such as adversarial robustness (Szegedy et al., 2014; Madry et al., 2018; Weng et al., 2018b), regularization (Arjovsky et al., 2017; Gulrajani et al., 2017), or theoretical bounds without practical considerations (Bartlett et al., 2017). The absence of a comprehensive survey integrating these perspectives leaves a critical gap in the literature. To address this, we provide a systematic review of Lipschitz continuity in deep learning, covering its **theoretical foundations**, **estimation methods**, **regularization approaches in optimization and architecture**, and **certifiable robustness**. By synthesizing key findings across these domains, this survey serves as a valuable resource for researchers and practitioners seeking a deeper understanding of Lipschitz continuity and its role in trustworthy learning systems. Beyond conducting a survey, we also critically distill the existing knowledge, present it in a more accessible manner, and provide the results or proofs **missing** or **incorrect** in the literature. The message of this survey is that Lipschitz continuity is not just a niche mathematical property; it is a fundamental principle for building and analyzing trustworthy neural networks.

## 1.1 Scope

This survey aims to consolidate existing knowledge across these domains, providing a comprehensive resource for researchers and practitioners seeking to understand and leverage Lipschitz continuity in deep learning. To ensure a high-quality review, we focus primarily on the literature published in prestigious and other well-regarded sources. The scope of this survey is organized as below:

1. Section 2: Theoretical Foundations presents the formal definitions and fundamental properties of Lipschitz continuity, followed by key theoretical developments in deep learning. It covers the mathematical foundations, Lipschitz properties of activation functions and neural networks, Lipschitz analysis of multi-head self-attention, complexity-theoretic generalization bounds, and advances in training dynamics.

2. Section 3: Estimation Methods surveys the principal methods for estimating the Lipschitz constants of neural networks, including derivative bound propagation, spectral alignment, convex optimization relaxations, and relaxed integer programming approaches.

3. Section 4: Regularization Approaches reviews approaches for enforcing Lipschitz continuity in neural networks through both optimization-based and architectural approaches, covering weight regularization, gradient regularization, activation regularization, and class-margin regularization.

4. Section 5: Certifiable Robustness examines methods for achieving certifiable robustness through Lipschitz bounds, including the theoretical foundations of Lipschitz-margin robustness radii, certification via global Lipschitz bounds, certification via local Lipschitz bounds and certificated robustness in LLMs.

## 1.2 Contributions

Our key contributions are as follows:

1. **A Unified Systematic Review**. We close a critical gap in the literature by conducting a comprehensive survey of Lipschitz continuity in deep learning. Our review unites disparate research threads

— spanning theoretical foundations, estimation methods, regularization approaches, architectural designs, and certified robustness — into a coherent framework that highlights their interconnections from the lens of Lipschitz continuity.

2. **Comprehensive Analysis**. We provide in-depth examinations of existing techniques organized into four categories:

   - **Theoretical Foundations**. We review the formal definitions and properties of Lipschitz continuity, analyze Lipschitz behavior of activations and network architectures, and synthesize key results on generalization bounds, optimization convergence, and training-induced Lipschitz dynamics.
   - **Estimation Methods**. From power-iteration spectral-norm approximations and extreme-value theory to randomized and interval-analysis approaches, we compare each method's tightness, computational overhead, and scalability to modern architectures.
   - **Regularization Approaches**. We critically assess gradient-based penalties, spectral normalization, orthogonal and Parseval constraints, and novel Lipschitz-bounded activations, examining their theoretical guarantees, implementation trade-offs, and empirical effects on robustness, generalization, and training dynamics.
   - **Certifiable Robustness**. We survey Lipschitz-based certification frameworks — covering global and local Lipschitz bounds, path-based and convex-relaxation certificates, randomized smoothing, and Lipschitz-constrained architectures — analyzing their provable guarantees, computational requirements, and applicability to real-world deep models.

3. **Results and Proofs**. We contribute new and complementary results, including corrected Lipschitz constants for common activation functions (validated by our numerical experiments) and a Lipschitz continuity bound for arbitrary neural networks represented as directed acyclic graphs (DAGs). Furthermore, we provide theoretical results **missing or incorrect** from prior work, such as rigorous proofs of the closed-form expressions for the Lipschitz constants of common activation functions and a general perturbation bound for certifiable robustness based on Hölder's conjugacy relation.

## 2 Theoretical Foundations

Lipschitz continuity is a foundational concept in deep learning theory, offering a rigorous framework for quantifying a neural network's sensitivity to input perturbations. It plays a central role in understanding robustness, generalization, and optimization dynamics. Throughout, we use operator $\mathrm{Lip}_p[\bullet]$ to denote the $p$-norm Lipschitz constant for $\bullet$, and use operator $\mathrm{Lip}[\bullet]$ to denote the 2-norm Lipschitz constant for $\bullet$.

### 2.1 Preliminaries

**Definition 2.1** ($p$-Norm in Finite-Dimensional Vector Spaces)**.** Let $V \subseteq \mathbb{R}^d$ be a finite-dimensional vector space. The $p$-norm (*i.e.* $\ell_p$-norm) in $V$ is defined as:

$$\|z\|_p = \begin{cases} \left(\sum\limits_{i=1}^{d} |z_i|^p\right)^{1/p}, & \text{if } 1 \le p < \infty, \\[2ex] \max\limits_{i=1,\dots,d} |z_i|, & \text{if } p = \infty, \end{cases} \tag{1}$$

where $z = (z_1, z_2, \dots, z_d) \in V$ (Rudin, 1987, Definition 3.6, § 3).

**Definition 2.2** (Globally $K$-Lipschitz Continuous)**.** Let $f : X \mapsto Y$ be a function, where $X \subseteq \mathbb{R}^d$ and $Y \subseteq \mathbb{R}^c$. The function $f$ is said to be *globally $K$-Lipschitz continuous* (with respect to the 2-norm) if there exists a constant scalar $K > 0$ such that:

$$\|f(u) - f(v)\|_2 \le K\|u - v\|_2, \quad \forall u, v \in X, \tag{2}$$

(Rudin, 1976, Theorem 9.19, § 9). Throughout, unless specified otherwise, the *Lipschitz norm* is assumed to be the 2-norm.

**Definition 2.3** (Globally $K_{p \to q}$-Lipschitz Continuous)**.** Let:

$$1 \leq p, q \leq \infty$$

be two scalars. A function $f : X \to Y$, with $X \subseteq \mathbb{R}^d$ and $Y \subseteq \mathbb{R}^c$, is said to be $K_{p \to q}$-*Lipschitz continuous* if there exists a constant $K_{p \to q} > 0$ such that:

$$\|f(u) - f(v)\|_q \ \leq \ K_{p \to q} \|u - v\|_p, \quad \forall u, v \in X, \tag{3}$$

where $\| \cdot \|_p$ and $\| \cdot \|_q$ denote the $\ell_p$ and $\ell_q$ norms, respectively. Particularly, $K_{2 \to 2}$-Lipschitz continuous reduces to $K$-Lipschitz continuous with respect to 2-norm.

*Remark* 2.4. Definition 2.3 (Globally $K_{p \to q}$-Lipschitz Continuous) is non-standard in the mathematical literature; however, it is useful for discussions of certifiable robustness in the context of deep learning.

**Lemma 2.5** (Lipschitz Constant Bounds Gradient Norm)**.** *Let $f : \mathbb{R}^m \to \mathbb{R}$ be a differentiable function. Then $f$ is $K$-Lipschitz continuous (with respect to the 2-norm) if and only if*

$$K \geq \sup_{x \in \operatorname{dom}(f)} \|\nabla f(x)\|_2, \tag{4}$$

*(Rudin, 1976, Theorem 9.19, § 9). In the case $0 < K < 1$, the function $f$ is called a contraction map. In particular, for the case that $\operatorname{dom}(f)$ is a convex set — any points connected through the* Mean Value Theorem *remain within $\operatorname{dom}(f)$, thus the tight bound is achieved such that:*

$$K = \sup_{x \in \operatorname{dom}(f)} \|\nabla f(x)\|_2. \tag{5}$$

*Remark* 2.6. In Lemma 2.5 (Lipschitz Constant Bounds Gradient Norm), of the deep learning sense, the domain of a neural network $f : \mathbb{R}^m \to \mathbb{R}$ can be assumed to be a convex set on $\mathbb{R}^m$. Consequently, the tight Lipschitz bound is attained.

## 2.2   1-**Lipschitz Orthogonal Matrix**

**Definition 2.7** (Skew-Hermitian & Skew-Symmetric Matrix)**.** A matrix $A$ is referred to as a *skew-hermitian matrix*, if $A$ satisfies:

$$A = -A^*. \tag{6}$$

Particularly, if $A$ is a real matrix, then $A$ is also referred to as a *skew-symmetric matrix*.

**Proposition 2.8** (Lipschitz Constant of Semi-Orthogonal Matrix)**.** *If a matrix $W \in \mathbb{R}^{m \times n}$ with:*

$$W^\top W = I_n \qquad or \qquad WW^\top = I_m, \tag{7}$$

*then $W$ is semi-orthogonal with Lipschitz constant 1. If $W^\top W = I_n$ (requiring $m \geq n$), $W$ is an isometric map over the entire $\mathbb{R}^n$; if $WW^\top = I_m$ (requiring $m \leq n$), $W$ is non-expansive, acting as an isometry on $\ker(W)^\perp$ and vanishing on $\ker(W)$.*

*Proof.*   **Case 1:** $m \geq n$ **(Isometric Map).** For $\forall\, x, y \in \mathbb{R}^n$:

$$\begin{aligned}
\|W(x - y)\|_2^2 &= (W(x - y))^\top W(x - y) \\
&= (x - y)^\top W^\top W(x - y) \\
&= (x - y)^\top I_n (x - y) \\
&= \|x - y\|_2^2,
\end{aligned}$$

then by definition:

$$\operatorname{Lip}[W] = \sup_{x \neq y} \frac{\|W(x - y)\|_2}{\|x - y\|_2} = 1.$$

**Case 2:** $WW^\top = I_m$, $m \leq n$ **(Vanishing on Kernel).** Since $WW^\top = I_m$:

$$\|W\|_2^2 = \lambda_{\max}(W^\top W) = \lambda_{\max}(WW^\top) = \lambda_{\max}(I_m) = 1,$$

so $\mathrm{Lip}\,[W] = \|W\|_2 = 1$. When $m < n$, $\ker(W)$ is non-trivial, and for $\exists\, x - y \in \ker(W)$:

$$\|W(x-y)\|_2 = 0 < \|x-y\|_2,$$

so $W$ vanishes on $\ker(W)$ and is thus non-expansive but not an isometry.

$\square$

### 2.2.1 Matrix Orthogonalization

We also survey useful matrix orthogonalization algorithms below.

**Lie Group Exponential Map.** Let:

$$U(n) = \left\{ A \in \mathbb{C}^{n \times n} \mid A^* A = I \right\} \tag{8}$$

be a unitary group and:

$$\mathfrak{u}(n) = \{ B \in \mathbb{R}^{n \times n} \; : \; B = -B^* \} \tag{9}$$

be a *Skew-Hermitian* group. Then there exists an *Lie Exponential Map*:

$$\exp(A) : \mathfrak{u}(n) \to U(n), \tag{10}$$

connecting the groups $\mathfrak{u}(n)$ and $U(n)$, defined as:

$$\exp(A) = I + A + \frac{1}{2}A^2 + \cdots, \tag{11}$$

which is surjective. For a matrix $W$, its corresponding skew-Hermitian matrix can be constructed by taking:

$$B = W - W^* \in \mathfrak{u}(n) \tag{12}$$

(Lezcano-Casado & Martínez-Rubio, 2019).

*Remark* 2.9. The *Lie Exponential Map* (equation 11) together with the skew-Hermitian matrix conversion (equation 12) allow us to map an arbitrary matrix $W$ into the unitary algebra $\mathfrak{u}(n)$ which has Lipschitz constant one. This is particularly useful for architecturally designing 1-Lipschitz neural networks.

**Cayley Transform.** If $B \in \mathbb{R}^{n \times n}$ is a *skew-symmetric* matrix, the map:

$$\phi(B) = (I + \frac{1}{2}B)(I - \frac{1}{2}B)^{-1} \tag{13}$$

is referred to as the *Cayley map* which transforms $B$ to an orthogonal matrix $\phi(B)$ (Cayley, 1846; Trockman & Kolter, 2021). Then the transformed orthogonal matrix $\phi(B)$ has a constant Lipschitz one. The Cayley map is not surjective onto the orthogonal group: its image excludes orthogonal matrices with $-1$ in their spectrum.

***Björck Orthogonalization* Approximation.** Suppose $W_0 \in \mathbb{R}^{n \times n}$ is not an orthogonal matrix. Let $W_k$ be an iterative sequence such that $W_k$ is the closest orthogonal matrix to $W_0$ as $k \to \infty$. *Björck Orthogonalization* (Björck & Bowie, 1971) is a differentiable method to iteratively solve the closest orthogonal matrix to a given matrix $W_0$. Chernodub & Nowicki (2017) use *Björck Orthogonalization* in an optimization problem for constraining the orthogonality of parameter matrices (Chernodub & Nowicki, 2017). *Björck Orthogonalization* (Björck & Bowie, 1971) is defined by:

$$W_{k+1} = W_k \left( I + \frac{1}{2}Q_k + \cdots + (-1)^p \binom{-\frac{1}{2}}{p} Q_k^p \right), \tag{14}$$

where $W_k$ is the $k$-th iterative result, $p$ is a chosen hyper-parameter, $\begin{pmatrix} -\frac{1}{2} \\ p \end{pmatrix}$ is binomial coefficient, and $Q_k$ is:

$$Q_k = I - W_k^\top W_k. \tag{15}$$

As $k \to \infty$, $W_k^\top W_k$ converges to $I_n$. For a rectangular matrix $W_0 \in \mathbb{R}^{m \times n}$, if $\mathrm{rank}(W_0) = n$, the convergence still holds:

$$W_k^\top W_k \to I_n,$$

as $k \to \infty$.

### 2.3 Functional Inequalities

**Proposition 2.10** (Lipschitz Constant of Function Composition)**.** *Let $f : X \to Y$ and $g : Y \to Z$ be two bounded functions. Then the Lipschitz constant of their composition $g \circ f : X \to Z$ admits:*

$$\mathrm{Lip}_p [g \circ f] \le \mathrm{Lip}_p [g] \ \mathrm{Lip}_p [f]. \tag{16}$$

*Proof.*

$$
\begin{aligned}
\mathrm{Lip}_p [g \circ f] &= \sup_{\|x-y\|_p \neq 0} \frac{\|g(f(x)) - g(f(y))\|_p}{\|x - y\|_p} \\
&\le \mathrm{Lip}_p [g] \sup_{\|x-y\|_p \neq 0} \frac{\|f(x) - f(y)\|_p}{\|x - y\|_p} \\
&= \mathrm{Lip}_p [g] \ \mathrm{Lip}_p [f].
\end{aligned}
$$

$\square$

**Proposition 2.11** (Lipschitz Constant of Function Addition)**.** *Let $f$ and $g$ be two bounded functions on $\mathrm{dom}(g) \cap \mathrm{dom}(f)$. Then the Lipschitz constant of $g + f$ admits:*

$$\mathrm{Lip}_p [g + f] \le \mathrm{Lip}_p [g] + \mathrm{Lip}_p [f]. \tag{17}$$

*Proof.*

$$
\begin{aligned}
\mathrm{Lip}_p [g + f] &= \sup_{\|x-y\|_p \neq 0} \frac{\|f(x) + g(x) - [f(y) + g(y)]\|_p}{\|x - y\|_p} \\
&= \sup_{\|x-y\|_p \neq 0} \frac{\|f(x) - f(y) + g(x) - g(y)\|_p}{\|x - y\|_p} \\
&\le \sup_{\|x-y\|_p \neq 0} \frac{\|f(x) - f(y)\|_p}{\|x - y\|_p} + \sup_{\|x-y\|_p \neq 0} \frac{\|g(x) - g(y)\|_p}{\|x - y\|_p} \\
&= \mathrm{Lip}_p [g] + \mathrm{Lip}_p [f].
\end{aligned}
$$

$\square$

**Proposition 2.12** (Lipschitz Constant of Function Concatenation)**.** *Let $f : \mathbb{R}^d \to \mathbb{R}^m$ and $g : \mathbb{R}^d \to \mathbb{R}^n$ be two Lipschitz continuous functions. Let:*

$$(f \oplus g)(x) := \begin{bmatrix} f(x) \\ g(x) \end{bmatrix} \in \mathbb{R}^{m+n} \tag{18}$$

*be a function by concatenating the outputs of $f$ and $g$. Then, for $1 \le p < \infty$, the following inequality holds true:*

$$\left( \mathrm{Lip}_p [f \oplus g] \right)^p \le \left( \mathrm{Lip}_p [f] \right)^p + \left( \mathrm{Lip}_p [g] \right)^p. \tag{19}$$

*Proof.* For any $x, y \in \mathbb{R}^d$, we have:

$$\left\| (f \oplus g)(x) - (f \oplus g)(y) \right\|_p^p = \left\| \begin{bmatrix} f(x) \\ g(x) \end{bmatrix} - \begin{bmatrix} f(y) \\ g(y) \end{bmatrix} \right\|_p^p$$

$$= \sum_{i=1}^{m} \left( |f(x) - f(y)|^p \right)_i + \sum_{j=1}^{n} \left( |g(x) - g(y)|^p \right)_j$$

$$\leq \left( \left( \operatorname{Lip}_p [f] \right)^p + \left( \operatorname{Lip}_p [g] \right)^p \right) \|x - y\|_p^p.$$

Hence:

$$\left( \operatorname{Lip}_p [f \oplus g] \right)^p = \sup_{x \neq y} \frac{\left\| (f \oplus g)(x) - (f \oplus g)(y) \right\|_p^p}{\|x - y\|_p^p}$$

$$\leq \left( \operatorname{Lip}_p [f] \right)^p + \left( \operatorname{Lip}_p [g] \right)^p.$$

$\square$

## 2.4 Sub-Differential Convex Functions

For a function that is not differentiable but is locally Lipschitz continuous — such as the ReLU activation — the Clarke sub-differential provides a generalized notion of gradient that allows one to analyze and bound local Lipschitz constants (Clarke, 1975).

**Definition 2.13** (Generalized Gradient). Let $f : \mathbb{R}^m \to \mathbb{R}$ be a function on $X$. The sub-differential of $f$ at a point $x$ is the set

$$\partial_s f(x) = \{ g \in \mathbb{R}^m \mid f(y) \geq f(x) + \langle g, y - x \rangle, \; \forall y \in X \}. \tag{20}$$

If $f$ is convex, then:

$$\partial_s f(x) \neq \varnothing \qquad \text{and} \qquad \partial_s f(x) = \{ \nabla f(x^*) \mid \forall \; x^* \to x \}. \tag{21}$$

**Definition 2.14** (Clarke Sub-Gradient). Let $f : \mathbb{R}^m \to \mathbb{R}$ be a proper convex function on $X$. If $f$ is locally Lipschitz continuous on $X$, then by Rademacher's theorem it is differentiable almost everywhere. The Clarke sub-differential $f$ at a point $x$ is defined as the convex hull of all limit points of gradients at differentiable points approaching $x$. That is, a vector $g$ lies in the Clarke sub-differential at $x$ if there exists a sequence of points $x_k \to x$ ($k \in \mathbb{N}$) at which $f$ is differentiable such that the gradients $\nabla f(x_k)$ converge to $g$. Then the Clarke sub-differential is a convex hull, satisfying:

$$\partial f(x) = \operatorname{conv}\left\{ g \mid \exists \text{ a sequence } (x_k) \to x \text{ as } k \to \infty, \nabla f(x_k) \to g \right\} \tag{22}$$

(Clarke, 1975, Definition 1.1). If $f$ is convex, then $\partial f(x) = \partial_s f(x)$. This definition also extends to vector-valued functions (Clarke, 1990).

**Lemma 2.15** (Local Lipschitz Constant of Sub-Differential Function). *Let $f : \mathbb{R}^m \to \mathbb{R}^n$ be a convex function differentiable almost everywhere on $X$. Then the $p \to q$ local Lipschitz constant of $f$ on $X$ is given by:*

$$\operatorname{Lip}_{p \to q}[f; X] = \sup_{x \in X} \sup_{G \in \partial f(x)} \sup_{\|v\|_p \neq 0} \frac{\|G^\top v\|_q}{\|v\|_p} = \sup_{x \in X} \sup_{G \in \partial f(x)} \sup_{\|v\|_p = 1} \|G^\top v\|_q, \tag{23}$$

*such that:*

$$\|f(x) - f(y)\|_q \leq \operatorname{Lip}_{p \to q}[f; X] \|x - y\|_p, \tag{24}$$

*for all $x, y \in X$ in the sense of Clarke sub-gradient (Jordan & Dimakis, 2020; Boyd et al., 2022).*

### 2.5 Lipschitz Continuity of Activation Functions

An activation function:

$$\rho : \mathbb{R}^d \to \mathbb{R}^d$$

is a nonlinear mapping applied after neuron's output at a layer, enabling neural networks to learn complex patterns. For example, if $\sigma$ is piecewise defined as:

$$\rho(x) = \begin{cases} x, & \text{if } x > 0, \\ 0, & \text{otherwise} \end{cases},$$

then the $\rho$ is referred to as *ReLU* activation function (Nair & Hinton, 2010).

The Lipschitz constants of commonly used activation functions are provided in Table 1. The constants are derived as the supremum of the 2-norm of the derivatives of activation functions. Proofs are provided in the Appendix A. To numerically validate the results, we have conducted experiments by maximizing the gradient norm using Lemma 2.5 for a further sanity check.

**Notes.** The Lipschitz constant for the *softmax* function reported in the literature (Gouk et al., 2021) and *Sigmoid* function reported in the literature (Virmaux & Scaman, 2018) as 1. We show in Appendix A.6 that the exact Lipschitz constant of *softmax* is indeed $\frac{1}{2}$, validated by our numerical experiment. This result has been reported in recent concurrent literature (Nair, 2025). We also show in Appendix A.1 that exact Lipschitz constant of *sigmoid* is indeed $\frac{1}{4}$, validated by our numerical experiment. For *Leaky ReLU* and *ELU*, the parameter $\alpha$ is typically chosen with $\alpha < 1$, so their effective Lipschitz constants are usually 1 in practice.

**Auto-Differentiation based Numerical Method.** Suppose $f : \mathbb{R}^d \to \mathbb{R}$ is a first- and second-order differentiable neural network under domain convex condition (Remark 2.6). Leveraging the autograd mechanism of PyTorch, we use gradient descent — *e.g.*, Adam (Kingma & Ba, 2014) — to numerically approximate the supremum:

$$\text{Lip}[f] = \sup_x \|f'(x)\|_2,$$

using Lemma 2.5.

Table 1: Lipschitz constants of activation functions.

| Activation | Definition | Lipschitz Constant & Proof | Citations |
|---|---|---|---|
| ReLU | $\max(0, x)$ | 1 | Nair & Hinton (2010, Sec. 2); Virmaux & Scaman (2018, Sec. 5) |
| Leaky ReLU | $\max(\alpha x, x)$ | $\max(1, \alpha)$ | Maas et al. (2013, Sec. 2); Virmaux & Scaman (2018, Sec. 5) |
| Sigmoid | $\frac{1}{1+e^{-x}}$ | $\frac{1}{4}$ (Appendix A.1) | Virmaux & Scaman (2018, Sec. 5) |
| Tanh | $\tanh(x)$ | 1 (Appendix A.2) | Cybenko (1989); Virmaux & Scaman (2018, Sec. 5) |
| Softplus | $\log(1 + e^x)$ | 1 (Appendix A.3) | Dugas et al. (2000, Sec. 2); Virmaux & Scaman (2018, Sec. 5) |
| ELU | $\begin{cases} x & x > 0 \\ \alpha(e^x - 1) & x \leq 0 \end{cases}$ | $\max(1, \alpha)$ | Clevert et al. (2016, Sec. 3) |
| Swish | $x \cdot \sigma(x)$ | $\approx 1.1$ (Appendix A.4) | Ramachandran et al. (2018, Sec. 1) |
| GELU | $\frac{x}{2}\left(1 + \text{erf}\left(\frac{x}{\sqrt{2}}\right)\right)$ | $\approx 1.13$ (Appendix A.5) | Hendrycks & Gimpel (2016, Sec. 2) |
| Softmax | $\text{softmax}(x_i) = \frac{e^{x_i}}{\sum_j e^{x_j}}$ | $\frac{1}{2}$ (Appendix A.6) | Gouk et al. (2021, Sec. 3.3) |

### 2.6 Lipschitz Continuity of Dot-Product Self-Attention

LLMs (Brown et al., 2020; Chowdhery et al., 2023; Touvron et al., 2023; OpenAI et al., 2024; DeepSeek-AI et al., 2025), built on transformer architectures (Vaswani et al., 2017), rely heavily on dot-product

self-attention mechanisms, whose Lipschitz continuity governs their sensitivity to input perturbations, such as token-level adversarial attacks. Dot-product self-attention mechanisms form the core of transformer architectures, enabling effective modeling of long-range dependencies in sequences for applications such as natural language processing and vision-language tasks.

**Single-Head Dot-Product Self-Attention.** For an input sequence matrix $x \in \mathbb{R}^{n \times d}$, where $n$ is the sequence length and $d$ is the input dimension, a self-attention layer computes the output as

$$\text{Attention}(x) = \underbrace{\text{softmax}\left(\frac{QK^\top}{\sqrt{d_K}}\right)}_{\text{row-wise softmax}} V \in \mathbb{R}^{n \times m}, \tag{25}$$

(Vaswani et al., 2017, Equation 1) where

$$Q = xW_Q \in \mathbb{R}^{n \times m}, \quad K = xW_K \in \mathbb{R}^{n \times m}, \quad V = xW_V \in \mathbb{R}^{n \times m}, \tag{26}$$

are the *query*, *key*, and *value* matrices, with weight matrices:

$$W_Q, W_K, W_V \in \mathbb{R}^{d \times d_K},$$

and $d_K$ denotes the key (and query) vector dimension. The softmax is applied row-wise to normalize attention scores. The scaling factor $\frac{1}{\sqrt{d_K}}$ stabilizes the computation numerically.

**Multi-Head Dot-Product Self-Attention.** In practice, transformers employ $h$ parallel self-attention heads to jointly attend to information from different representation subspaces (Vaswani et al., 2017). Each head $i$ has its own projection matrices:

$$W_{Q,i}, W_{K,i}, W_{V,i} \in \mathbb{R}^{d \times m_i},$$

where $m_i = \frac{m}{h}$, $h$ denotes the number of heads, and produces an output for head $i$:

$$\text{Attention}_i(x) \in \mathbb{R}^{n \times m_i}. \tag{27}$$

The outputs are concatenated:

$$\text{Attention}(x) = \big[\text{Attention}_1(x), \text{Attention}_2(x), \cdots, \text{Attention}_h(x)\big] \in \mathbb{R}^{n \times m}, \tag{28}$$

produces the final output.

**Locally Lipschitz Continuous.** Standard dot-product self-attention is locally Lipschitz continuous, and several works have derived explicit upper bounds for its local Lipschitz constant. Let $\mathcal{B}(x_0, \delta) = \{x \mid \|x - x_0\|_2 < \delta\}$ denote a ball centered $x_0$ with a radius $\delta$. Hu et al. show that for the dot-product self-attention layer within a ball $\mathcal{B}_2(x, \delta)$ is locally Lipschitz continuous bounded by:

$$\text{Lip}\Big[\text{Attention}; \mathcal{B}_2(x, \delta)\Big] \leq n(n+1)\Big(\|x\|_2 + \delta\Big)^2 \Big[\|W_V\|_2\|W_Q\|_2\|W_K^\top\|_2 + \|W_V\|_2\Big], \tag{29}$$

where $x \in \mathbb{R}^{n \times d}$, $n$ is the sequence length, $d$ is the input dimension, $W_K, W_Q, W_V \in \mathbb{R}^{d \times d_K}$ are the *key*, *query* and *value* parameter matrices, and $\delta$ is a 2-norm radius (Hu et al., 2024, Theorem 2). Castin et al. provide a tighter bound by $\sqrt{n}$ up to a constant factor, showing that within a ball $\mathcal{B}_2(0, \delta)$, the Lipschitz constant of dot-product self-attention is bounded by:

$$\text{Lip}\Big[\text{Attention}; \mathcal{B}_2(0, \delta)\Big] \leq \sqrt{3}\|W_V\|_2 \left[\left\|\frac{W_K^\top W_Q}{\sqrt{d_K}}\right\|_2^2 \delta^4(4n+1) + n\right]^{\frac{1}{2}}, \tag{30}$$

where $A$ is the attention score matrix (Castin et al., 2024, Theorem 3.3).

Most recently, Yudin et al. (2025) refine the bound by explicitly incorporating the Jacobian of the softmax function, given by:

$$\mathcal{M}(P_x) := \nabla \operatorname{softmax}(x) = \operatorname{diag}(P_x) - P_x P_x^\top, \tag{31}$$

where:

$$P_x = \frac{e^{x_i}}{\sum_j e^{x_j}},$$

showing that the Lipschitz constant for a single head is bounded by:

$$\operatorname{Lip}\Big[\operatorname{Attention}\Big] \leq \|W_V\|_2 \left( \|P\|_2 + 2\|x\|_2^2 \, \|A\|_2 \, \max_i \|\mathcal{M}(P_{i,:})\|_2 \right) \tag{32}$$

with:

$$P = \operatorname{softmax}\Big(xAx^\top\Big) \in \mathbb{R}^{n \times n} \qquad \text{and} \qquad A = \frac{W_Q W_K^\top}{\sqrt{d_K}} \in \mathbb{R}^{d \times d}$$

(Yudin et al., 2025, Theorem 3).

**Globally Non-Lipschitz Continuous.** It is worth noting that the Lipschitz constants of both single-head and multi-head dot-product self-attention are usually not globally bounded for arbitrary inputs. This is because the Lipschitz bound for dot-product self-attention grows with the sequence length and input norm itself, implying that the dot-product self-attention is not globally Lipschitz continuous (Kim et al., 2021, Theorem 3.1).

## 2.7 Lipschitz Continuity of Neural Networks

**Definition 2.16** (Feedforward Network (Compositional Definition)). Let $f : \mathbb{R}^m \to \mathbb{R}^n$ denote a feedforward neural network consisting of $L$ layers, defined as the composition:

$$f := h^{(L)} \circ \cdots \circ h^{(2)} \circ h^{(1)},$$

where each layer $h^{(\ell)}$ is composed of a linear transformation $\phi^{(\ell)}(\cdot)$ followed by a non-linear activation function $\rho^{(\ell)}(\cdot)$:

$$h^{(\ell)} := \rho^{(\ell)} \circ \phi^{(\ell)}.$$

**Proposition 2.17** (Lipschitz Constant of a Linear Layer). *Let $\phi^{(\ell)} : \mathbb{R}^{m_\ell} \to \mathbb{R}^{n_\ell}$ denote a linear transformation implemented either as a convolutional layer:*

$$\phi^{(\ell)}(z^{(\ell-1)}) = \boldsymbol{\theta}^{(\ell)} \circledast z^{(\ell-1)} + b^{(\ell)},$$

*or as a fully connected (dense) layer:*

$$\phi^{(\ell)}(z^{(\ell-1)}) = \boldsymbol{\theta}^{(\ell)} z^{(\ell-1)} + b^{(\ell)},$$

*where $\boldsymbol{\theta}^{(\ell)}$ is the weight matrix (or convolution kernel), $b^{(\ell)}$ is the bias term, $\circledast$ denotes the convolution operator, and $z^{(\ell-1)}$ is the output from the previous layer. The Lipschitz constant of $\phi^{(\ell)}$ with respect to the Euclidean norm is given by:*

$$\operatorname{Lip}[\phi^{(\ell)}] = \|\boldsymbol{\theta}^{(\ell)}\|_2,$$

*where $\|\boldsymbol{\theta}^{(\ell)}\|_2$ denotes the spectral-norm of the associated linear operator.*

**Proposition 2.18** (Spectral-Norm via Truncated Singular Value Decomposition). *Let $\boldsymbol{\theta} \in \mathbb{R}^{n \times m}$ be a matrix, admitting a truncated singular value decomposition (SVD):*

$$\boldsymbol{\theta} = \sum_{i=1}^{r} \sigma_i \boldsymbol{u}_i \boldsymbol{v}_i^{\top}, \quad with \quad \sigma_1 > \sigma_2 > \cdots > \sigma_r > 0, \quad r = \operatorname{rank}(\boldsymbol{\theta}),$$

*where $\sigma_i$ are the singular values, and $\boldsymbol{u}_i$, $\boldsymbol{v}_i$ are the left and right singular vectors, respectively. Then, the spectral-norm of $\boldsymbol{\theta}$ (i.e., its operator norm induced by the Euclidean norm) is given by the largest singular value:*

$$\|\boldsymbol{\theta}\|_2 = \sigma_1$$

*(Horn & Johnson, 2012, Example 5.6.6, § 5).*

**Proposition 2.19** (Lipschitz Constant Upper Bound of Feedforward Neural Network). *Let $f : \mathbb{R}^m \to \mathbb{R}^n$ be an L-layer feedforward neural network composed of linear transformations $\phi^{(\ell)}$ and activation functions $\rho^{(\ell)}$. The Lipschitz constant of $f$ with respect to the Euclidean norm is defined as:*

$$\operatorname{Lip}[f] = \sup_{\forall \ u \neq v} \frac{\|f(u) - f(v)\|_2}{\|u - v\|_2}, \tag{33}$$

*which immediately admits an upper bound by applying Proposition 2.10 (Lipschitz Constant of Function Composition):*

$$\operatorname{Lip}[f] \leq \prod_{\ell=1}^{L} \operatorname{Lip}[\rho^{(\ell)}] \prod_{\ell=1}^{L} \operatorname{Lip}[\phi^{(\ell)}] \tag{34}$$

*(Luo et al., 2025a, Proposition 8, § 5).*

**Corollary 2.20** (Spectral-Norm Upper Bound of Feedforward Neural Network). *For an $\ell$-layer feedforward network $f$ with only linear or convolutional layers and 1-Lipschitz activation functions, combining Proposition 2.17 and Proposition 2.19 immediately yields a spectral-norm product bound:*

$$\operatorname{Lip}[f] \leq \prod_{\ell=1}^{L} \|\theta^{(\ell)}\|_2, \tag{35}$$

*where $\theta^{(\ell)}$ is the $\ell$-th layer parameter matrix.*

### 2.7.1 Lipschitz Continuity for a Directed Acyclic Graph (DAG)

To the best of our knowledge, the existing literature does not present an explicit Lipschitz bound for a general feedforward network with skip connections. Prior literature such as (Yoshida & Miyato, 2017; Virmaux & Scaman, 2018) focuses strictly on sequential architectures, while norm-based capacity measures such as the path norm (Neyshabur et al., 2015) implicitly involve path enumeration but are not formulated as direct Lipschitz bounds. More recent path-metric bounds (Gonon et al., 2025) apply to arbitrary DAGs but are expressed in a rescaling-invariant metric form rather than the simple sum-over-paths structure we consider.

Figure 1 provides an illustrative example of a DAG network. To complement the survey, we derive an explicit bound in Theorem 2.21 using graph-theoretic method. To the best of our knowledge, this result has not previously appeared in the literature.

**Theorem 2.21** (Lipschitz Bound for DAG Network). *Let $G = (V, E)$ be a finite directed acyclic graph (DAG) representing a feedforward neural network with unique input node $s \in V$ and output node $t \in V$. Each node $v \in V$ is a module $h_v$ that is Lipschitz continuous with constant $\operatorname{Lip}[h_v]$. An edge $(u \to v) \in E$ represents the computation of $h_v$ immediately after $h_u$. Let $x_{v_i} : x \to x_{v_i}(x)$ represent the output at a node $v_i \in V$. The network is evaluated additively along incoming edges:*

$$\begin{cases} x_s(x) = x \\ x_v(x) = \sum_{(u \to v) \in E} h_v\big(x_u(x)\big), & if \quad v \neq s \end{cases}. \tag{36}$$

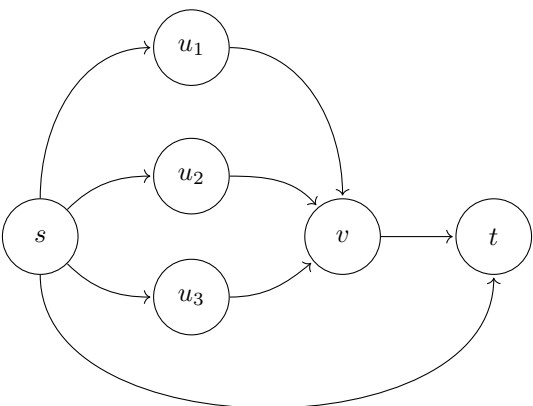

Figure 1: Example of Feedforward DAG Network. There are four computational paths: $s \to u_1 \to v \to t$, $s \to u_2 \to v \to t$, $s \to u_3 \to v \to t$, and $s \to t$. A node is a module in the DAG neural network $f$.

*A computational path $p$ from $s$ to $t$ is an ordered sequence of nodes*

$$p = (v_0 = s, v_1, \ldots, v_{L_p} = t),$$

*where $(v_{i-1} \to v_i) \in E$ for all $i = 1, \ldots, L_p$. Define for each edge $(u \to v)$ the constant*

$$C_{(u \to v)} := \mathrm{Lip}[h_v], \tag{37}$$

*and for a path $p$ set*

$$C_p := \prod_{i=1}^{L_p} \mathrm{Lip}[h_{v_i}] = \prod_{i=1}^{L_p} C_{(v_{i-1} \to v_i)}. \tag{38}$$

*Let $\mathcal{P}$ be the set of all such paths from $s$ to $t$. Then the Lipschitz constant of the overall network $f(x) := x_t(x)$ satisfies*

$$\mathrm{Lip}[f] \ \leq \ \sum_{p \in \mathcal{P}} C_p. \tag{39}$$

*Proof.* Fix any topological order of $V$. For $v \in V$, define the scalar

$$S(v) := \begin{cases} 1, & v = s, \\ \displaystyle\sum_{(u \to v) \in E} C_{(u \to v)} \, S(u), & v \neq s. \end{cases} \tag{40}$$

*Claim:* for all $v \in V$,

$$\mathrm{Lip}[x_v] \ \leq \ S(v). \tag{41}$$

We prove equation 41 by induction along the topological order.

**Base case $(v = s)$.** Considering $x_s(x) = x$, hence base case holds true:

$$\mathrm{Lip}[x_s] = 1 = S(s).$$

**Inductive step.** Let $v \neq s$ and assume equation 41 holds for all predecessors $u$ of $v$. Applying Minkowski inequality, for any $x, y$ in the input space,

$$
\begin{aligned}
\|x_v(x) - x_v(y)\| &= \left\| \sum_{(u \to v)} h_v\big(x_u(x)\big) - \sum_{(u \to v)} h_v\big(x_u(y)\big) \right\| \\
&\leq \sum_{(u \to v)} \left\| h_v\big(x_u(x)\big) - h_v\big(x_u(y)\big) \right\| \\
&\leq \sum_{(u \to v)} \mathrm{Lip}[h_v] \left\| x_u(x) - x_u(y) \right\| \\
&\leq \sum_{(u \to v)} \mathrm{Lip}[h_v] \, \mathrm{Lip}[x_u] \, \|x - y\| \\
&= \sum_{(u \to v)} C_{(u \to v)} \, \mathrm{Lip}[x_u] \, \|x - y\| \\
&\leq \left( \sum_{(u \to v)} C_{(u \to v)} \, S(u) \right) \|x - y\| = S(v) \, \|x - y\|,
\end{aligned}
$$

since inductive hypothesis:

$$
\mathrm{Lip}[x_u] \leq S(u),
$$

so that $\mathrm{Lip}[x_v] \leq S(v)$. In particular,

$$
\mathrm{Lip}[f] = \mathrm{Lip}[x_t] \ \leq \ S(t).
$$

It remains to show $S(t) = \sum_{p \in \mathcal{P}} C_p$. More generally:

**Lemma 2.22** (Lemma (Path expansion of $S$)). *For every $v \in V$, the path extension of $S$ holds:*

$$
S(v) = \sum_{p \in \mathcal{P}(s \to v)} \prod_{e \in p} C_e, \tag{42}
$$

*where $\mathcal{P}(s \to v)$ is the set of all directed paths from $s$ to $v$.*

*Proof.* Proceed again by induction in topological order. For $v = s$, $\mathcal{P}(s \to s)$ contains only the empty path with product 1, so equation 42 holds. Assume equation 42 holds for all predecessors $u$ of $v$. Then

$$
S(v) = \sum_{(u \to v)} C_{(u \to v)} \, S(u) \tag{43}
$$

$$
= \sum_{(u \to v)} C_{(u \to v)} \left( \sum_{p \in \mathcal{P}(s \to u)} \prod_{e \in p} C_e \right)
$$

$$
= \sum_{(u \to v)} \sum_{p \in \mathcal{P}(s \to u)} \left( \prod_{e \in p} C_e \right) C_{(u \to v)}.
$$

Concatenating each $p \in \mathcal{P}(s \to u)$ with the edge $(u \to v)$ yields a bijection onto $\mathcal{P}(s \to v)$, and multiplies the edge constants accordingly; thus equation 42 holds for $v$. $\quad\square$

Applying Lemma 2.22 at $v = t$ gives $S(t) = \sum_{p \in \mathcal{P}} C_p$, and hence

$$
\mathrm{Lip}[f] \ \leq \ S(t) = \sum_{p \in \mathcal{P}} \prod_{i=1}^{L_p} \mathrm{Lip}[h_{v_i}],
$$

which is the claim as demonstrated. $\quad\square$

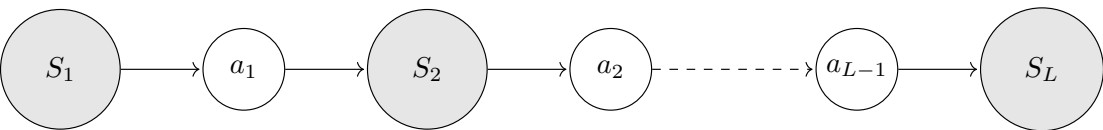

Figure 2: Topology of non-biconnected DAG network. If a DAG is separated into two sub-DAGs by the removal of a vertex $a_i$, then $a_i$ is referred to as a *cut vertex* (or *articulation point*), and the DAG is said to be *non-biconnected*. This diagram shows the topology for a DAG that contains $L-1$ articulation points $a_1, a_2, \ldots, a_{L-1}$ and $L$ corresponding sub-DAGs $S_1, S_2, \ldots, S_L$.

### 2.7.2 Lipschitz Constant of Non-Biconnected DAG

If a DAG is separated into two disconnected sub-DAGs by the removal of a vertex, then the removed vertex is referred to as *cut vertex* or *articulation point*. Modern deep learning architectures often give rise to computation DAGs that are not biconnected: the DAGs can be separated into multiple subgraphs by removing a small set of vertices.

**Theorem 2.23** (Lipschitz Bound for Non-Biconnected DAG Network). *Figure 2 shows the typical non-biconnected DAG topology found in modern deep learning architectures. The DAG contains a set of cut vertices $a_1, a_2, \cdots, a_{L-1}$ and biconnected DAGs $S_1, S_2, \cdots, S_L$. Suppose the DAG network $f$ in Figure 2 is:*

$$f = S_L \circ a_{L-1} \cdots a_2 \circ S_2 \circ a_1 \circ S_1,$$

*then it is not difficult to show that the Lipschitz constant bound of this DAG is:*

$$\mathrm{Lip}\,[f] \leq \left( \prod_{i=1}^{L} \mathrm{Lip}\,[S_i] \right) \left( \prod_{j=1}^{L-1} \mathrm{Lip}\,[a_j] \right), \tag{44}$$

*where the Lipschitz bound of each sub-DAG $\mathrm{Lip}\,[S_i]$ is given by Theorem 2.21 (Lipschitz Bound for DAG Network).*

*Remark* 2.24. Theorem 2.23 (Lipschitz Bound for Non-Biconnected DAG Network) is particularly valuable for estimating the Lipschitz bound of a deep model. In contrast to path-based bound of Theorem 2.21 (Lipschitz Bound for DAG Network), which can be quite loose due to its combinatorial dependence on the number of paths, Theorem 2.23 yields significantly tighter and more structurally informed bounds.

### 2.7.3 Lipschitz Constant of Residual Network

Residual networks (He et al., 2016) address the optimization challenges of very deep neural architectures by introducing *residual modules*, which are illustrated in Figure 3. Each module $m_{\mathrm{res}}$ has the structure:

$$m_{\mathrm{res}}(x) = x + \phi(x),$$

which consists of a non-identity unit $\phi$ and an identity skip connection. The resulting residual mapping $m_{\mathrm{res}}(x)$ maintains stable gradient flow, mitigates vanishing-gradient effects, and enables the training of substantially deeper networks without degradation. This simple architectural principle has proven highly effective and forms the backbone of many state-of-the-art deep learning models. Using Theorem 2.21 (Lipschitz Bound for DAG Network), it is not difficult to show that the Lipschitz constant bound of this structure is:

$$\mathrm{Lip}\,[m_{\mathrm{res}}] \leq \mathrm{Lip}\,[s] + \mathrm{Lip}\,[\phi] = 1 + \mathrm{Lip}\,[\phi],$$

which recovers the same results in (Behrmann et al., 2019, Lemma 2, § 2), (Gouk et al., 2021, § 3.4) and Proposition 2.10 (Lipschitz Constant of Function Composition) by setting $s$ to an identity map.

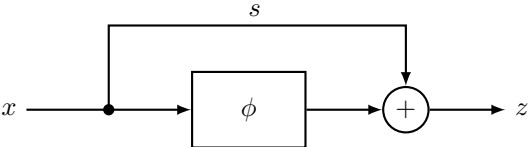

Figure 3: A residual module $m(x) = x + \phi(x)$ consists of a non-identity unit $\phi : x \mapsto \phi(x)$ and an identity skip connection unit $s : x \mapsto x$.

## 2.8 Complexity-Theoretic Generalization Bound

**Notations.** Let $\mathcal{H}$ be a hypothesis family and $\mathcal{H} \ni h : X \to Y$ be a hypothesis. Let $\ell : Y \times Y \to \mathbb{R}$ be a loss function. For each $\ell(h(x), y)$, we can associate $\ell$ and $h$ with a map $G \ni g : X \times Y \to \mathbb{R}$, where $G$ is the family of loss functions associated with the hypothesis family $\mathcal{H}$.

Lipschitz constants can enter complexity–theoretic generalization bounds and often interact with capacity, robustness, and smoothness (Mohri et al., 2018). To quantify this capacity for a hypothesis, the concept of *generalization gap* is often used to measure the error between *true risk* and *empirical risk* (Definition 2.25). This gap for a model $h \in \mathcal{H}$ is proportional to its empirical Rademacher complexity of the hypothesis family $\mathcal{H}$, and the empirical Rademacher complexity is proportional to the Lipschitz constant $\mathrm{Lip}\,[\mathcal{H}] := \sup_{h \in \mathcal{H}} \mathrm{Lip}\,[h]$ of the hypothesis family:

$$\text{Generalization gap of } h \sim \text{Rademacher complexity} \sim O(\mathrm{Lip}\,[\mathcal{H}])$$

(Mohri et al., 2018, § 3; Shalev-Shwartz & Ben-David, 2014, Part IV). A smaller Lipschitz constant enforces smoother mappings, reducing the complexity of the function class and mitigating overfitting, thereby enhancing robustness to noise and adversarial perturbations (Neyshabur et al., 2015; Gouk et al., 2021; Castin et al., 2024).

**Definition 2.25** (Generalization Gap). Let $S := \{(x_i, y_i)\}_{i=1}^m$ be a dataset where $(x_i, y_i)$ is sampled *i.i.d.* from an unknown distribution $\mathcal{D}_X$. The true (population) risk for hypothesis $h$ is defined as:

$$R(h) := \mathbb{E}_{(x,y) \sim \mathcal{D}_X}[\ell(h(x), y)],$$

and the empirical risk on $S$ is defined as:

$$R_S(h) := \frac{1}{m} \sum_{i=1}^m \ell(h(x_i), y_i)$$

(Mohri et al., 2018, § 3; Shalev-Shwartz & Ben-David, 2014, Part IV). Accordingly, the generalization gap on $S$ is given by:

$$R(h) - R_S(h).$$

### 2.8.1 Rademacher Complexity and its Bounds

In learning theory, *Rademacher complexity* quantifies the expressiveness of a hypothesis family $G := \{g : X \to \mathbb{R}\}$ by measuring how well the $G$ can fit data associated with random labels (Mohri et al., 2018, § 3.1). Let $S = (x_1, x_2, \ldots, x_m)$ be a dataset of size $m$ with each $x_i \in X$ drawn i.i.d. from an unknown distribution $\mathcal{D}_X$. Let $h : X \to \mathbb{R}$ be a hypothesis that assigns a real-valued score to each input. Let $\sigma_1, \ldots, \sigma_m$ be $m$ independent *Rademacher variables*, *i.e.*, random variables uniformly distributed over $\{-1, +1\}$ (Mohri et al., 2018, § 3.1). For random labels $\sigma_1, \ldots, \sigma_m$ assigned to $S$, the maximal correlation between the labels and the predictions of hypotheses in $\mathcal{H}$ is given by

$$\sup_{g \in G} \frac{1}{m} \sum_{i=1}^m \sigma_i g(x_i).$$

By assigning all possible random labels to the dataset $S$, we can thus define *Rademacher Complexity* for hypothesis family $\mathcal{G}$, stated in Definition 2.26 (Rademacher Complexity (Mohri et al., 2018, Definition 3.1, § 3)).

**Definition 2.26** (Rademacher Complexity (Mohri et al., 2018, Definition 3.1, § 3)). The *empirical Rademacher complexity* of $G$ with respect to $S$ is defined as

$$\widehat{\mathfrak{R}}_S(G) := \mathbb{E}_\sigma \left[ \sup_{g \in G} \frac{1}{m} \sum_{i=1}^m \sigma_i g(x_i) \right],$$

and the (expected) *Rademacher complexity* is the expectation of $\widehat{\mathfrak{R}}_S(G)$ over all dataset $S$ of size $m$:

$$\mathfrak{R}_m(G) := \mathbb{E}_{S \sim \mathcal{D}_X^m} \left[ \widehat{\mathfrak{R}}_S(G) \right],$$

where $S \sim \mathcal{D}_X^m$ denotes $m$ samples i.i.d. drawn from $\mathcal{D}$.

In the paradigm of machine learning, for the case of the hypothesis family $\ell \circ \mathcal{H}$ is the composition of a model $h \in \mathcal{H} := \{h : \mathbb{R}^n \to \mathbb{R}\}$ and loss function $\ell : \mathbb{R} \to \mathbb{R}$:

$$\ell \circ \mathcal{H} := \{\ell \circ h \mid h \in \mathcal{H}\},$$

by Talagrand's Contraction Lemma (Mohri et al., 2018, Lemma 4.2, § 4), the empirical Rademacher complexity of $\ell \circ \mathcal{H}$ on $S$ admits:

$$\widehat{\mathfrak{R}}_S(\ell \circ \mathcal{H}) \leq \operatorname{Lip}[\ell] \; \widehat{\mathfrak{R}}_S(\mathcal{H}).$$

For any $\delta > 0$, with probability at least $1 - \delta$, the generalization gap for $h \in \mathcal{H}$ on $S$ is bounded by empirical Rademacher complexity of $\ell \circ \mathcal{H}$:

$$R(h) - R_S(h) \leq 2\,\widehat{\mathfrak{R}}_S(\ell \circ \mathcal{H}) + 3\sqrt{\frac{\log \frac{2}{\delta}}{2m}} \leq 2\operatorname{Lip}[\ell]\;\widehat{\mathfrak{R}}_S(\mathcal{H}) + 3\sqrt{\frac{\log \frac{2}{\delta}}{2m}} \tag{45}$$

(Mohri et al., 2018, Theorem 3.1 & Theorem 3.3, § 3).

For general vectorized function, let:

$$\Phi := \{\phi : \mathbb{R}^n \to \mathbb{R}^n \mid \phi \text{ is a coordinate-wise identity map}\}$$

be a coordinate-wise identity family. For vectorized function $h : \mathbb{R}^n \to \mathbb{R}^k$ and compositional hypothesis $h \circ \Phi$, suppose $h_i$ is $K$-Lipschitz continuous, using vector contraction theorem (Maurer, 2016, Equation 1 & Corollary 4) yields:

$$\begin{aligned}
\widehat{\mathfrak{R}}_S(h \circ \Phi) &\leq \sqrt{2}\,K\,\frac{1}{m}\,\mathbb{E}\left[\sup_{\phi \in \Phi} \sum_{i=1}^m \sum_{j=1}^n \sigma_{ij}\phi_j(x_i)\right] \\
&\leq \sqrt{2}\,K\,\frac{1}{m}\,\sum_{i=1}^m \sum_{j=1}^n \left|\phi_j(x_i)\right| = \sqrt{2}\,K\,\frac{1}{m}\,\sum_{i=1}^m \left(\sum_{j=1}^n |x_{i:j}|\right) = \sqrt{2}\,K\,\frac{1}{m}\,\sum_{i=1}^m \|x_i\|_1 \\
&\leq \sqrt{2}\,K\,\frac{1}{m}\,m\,\sup_{x \in S}\|x\|_1 = \sqrt{2}\,K\,\sup_{x \in S}\|x\|_1,
\end{aligned} \tag{46}$$

where $\sigma_{ij}$ are an $m \times n$ matrix of independent Rademacher variables, $x_{i:j}$ denotes the $j$-th component of the $i$-th data point, and $\sup_{x \in S}\|x\|_1$ is the 1-norm dataset diameter. This result implies that Lipschitz constant of $h$ controls its generalization bound:

$$R(h) - R_S(h) \leq \mathcal{O}(K). \tag{47}$$

## 2.9 Training Dynamics of Lipschitz Continuity

The Lipschitz continuity of a parameterized network evolves as optimization updates its parameters. Consequently, the optimization process induces the dynamics of the network's Lipschitz continuity. While existing work in deep learning theory has largely focused on bounding the Lipschitz constants of neural networks, understanding how Lipschitz continuity evolves throughout training remains an open problem. Only a few studies have begun to explore this aspect. In particular, Luo et al. (2025a) introduces the concept of *optimization-induced dynamics* and establishes a continuous-time stochastic framework that explicitly links the optimization process to the time-varying Lipschitz continuity bound of deep networks. The literature (Luo et al., 2025a) uses the operator-theoretical results regarding how the largest singular values of parameter matrices vary with respect to perturbations from the literature (Luo et al., 2025b), along with the theory from high-dimensional stochastic differential equations (SDEs) from stochastic analysis, for establishing a mathematical framework modeling the evolution of a spectral-norm based Lipschitz continuity upper bound. They further validate their theoretical framework with experiments across datasets and regularization scenarios.

Consider an $L$-layer feed-forward neural network $f : \mathbb{R}^m \to \mathbb{R}$ with 1-Lipschitz activation functions (*e.g.*, ReLU). Let $\boldsymbol{\theta}^{(\ell)}(t) \in \mathbb{R}^{m_\ell \times n_\ell}$ be the $\ell$-layer parameter matrix at time $t$ and $\boldsymbol{\theta}(t)$ be the collection of $\boldsymbol{\theta}^{(\ell)}(t)$ for all layers. Let $\mathcal{L}_f(\boldsymbol{\theta}^{(\ell)}(t))$ be the loss expectation for the $f$ with parameters $\boldsymbol{\theta}(t)$ at time $t$. Suppose that $\boldsymbol{\Sigma}_t^{(\ell)} \in \mathbb{R}^{(m_\ell n_\ell) \times (m_\ell n_\ell)}$ is the layer-wise covariance matrix of stochastic gradient noise, arising from mini-batch sampling. Let $K^{(\ell)}(t)$ be the Lipschitz constant at layer $\ell$. Let $K(t)$ be the network Lipschitz spectral-norm bound. Luo et al. (2025a) show that the continuous-time dynamics of the parameters under stochastic gradient descent (SGD) with a sufficiently small learning rate $\eta > 0$ are given by a system of SDEs (Luo et al., 2025a, Definition 10):

$$\begin{cases} \mathrm{dvec}(\boldsymbol{\theta}^{(\ell)}(t)) = -\mathrm{vec}\left[\nabla^{(\ell)}\mathcal{L}_f(\boldsymbol{\theta}(t))\right]\,\mathrm{d}t + \sqrt{\eta}\left[\boldsymbol{\Sigma}_t^{(\ell)}\right]^{\frac{1}{2}}\,\mathrm{d}\boldsymbol{B}_t^{(\ell)} \\ K^{(\ell)}(t) = \|\boldsymbol{\theta}^{(\ell)}(t)\|_{op} \\ Z(t) = \sum_{l=1}^{L} \log K^{(\ell)}(t) \\ K(t) = \mathrm{e}^{Z(t)} \end{cases} \tag{48}$$

where:

- $\mathrm{vec} : \mathbb{R}^{m \times n} \to \mathbb{R}^{mn}$ represents major-column vectorization operator.

- $\|\cdot\|_{op}$ the matrix operator (spectral) norm.

- $\nabla^{(\ell)}\mathcal{L}_f(\boldsymbol{\theta}(t))$ is the gradient of the loss with respect to the $\ell$-th layer parameters.

- $\boldsymbol{B}_t^{(\ell)}$ is a standard $(m_\ell n_\ell)$-dimensional Wiener process adapted to the filtration induced by mini-batch sampling.

This stochastic dynamical system characterizes, in continuous time, the evolution of both layer-wise and network-level Lipschitz continuity bounds induced by the optimization process. Luo et al. (2025a, Theorem 15) show that the layer-wise dynamics decompose into three *driving forces*:

$$\frac{\mathrm{d}K^{(\ell)}(t)}{K^{(\ell)}(t)} = \left(\mu^{(\ell)}(t) + \kappa^{(\ell)}(t)\right)\,\mathrm{d}t + \boldsymbol{\lambda}^{(\ell)}(t)^\top \mathrm{d}\boldsymbol{B}_t^{(\ell)}, \tag{49}$$

where $\mathrm{d}\boldsymbol{B}_t^{(\ell)} \sim \mathcal{N}(\mathbf{0}, \mathbf{I}_{m_\ell n_\ell}\mathrm{d}t)$ represents the increment of a standard Wiener process in $\mathbb{R}^{m_\ell n_\ell}$, and the three *driving forces* are:

1. **Optimization-induced drift**:

$$\mu^{(\ell)}(t) = \frac{\left\langle \boldsymbol{J}_{op}^{(\ell)}(t), -\mathrm{vec}\left[\nabla^{(\ell)}\mathcal{L}_f(\boldsymbol{\theta}(t))\right] \right\rangle}{\sigma_1^{(\ell)}(t)}, \tag{50}$$

where $\boldsymbol{J}_{op}^{(\ell)}(t)$ is the Jacobian of the operator norm with respect to $\mathrm{vec}(\boldsymbol{\theta}^{(\ell)}(t))$ and $\sigma_1^{(\ell)}(t)$ is its largest singular value. This term captures the deterministic component of Lipschitz evolution driven by the mean gradient flow. This shows that the alignment of the gradient $\nabla^{(\ell)}\mathcal{L}_f(\boldsymbol{\theta}(t))$ and principal direction of the $\ell$-th layer parameter matrix $\boldsymbol{J}_{op}^{(\ell)}(t)$ at time $t$ determines the contribution of the optimization to the increment of the layer-wise Lipschitz constant.

2. **Noise–curvature entropy production**:

$$\kappa^{(\ell)}(t) = \frac{\eta}{2\,\sigma_1^{(\ell)}(t)} \left\langle \boldsymbol{H}_{op}^{(\ell)}(t), \boldsymbol{\Sigma}_t^{(\ell)} \right\rangle \geq 0, \tag{51}$$

where $\boldsymbol{H}_{op}^{(\ell)}(t)$ is the Hessian of the operator norm and $\boldsymbol{\Sigma}_t^{(\ell)}$ is the gradient-noise covariance. This non-negative term quantifies irreversible growth of the Lipschitz bound due to the interaction between stochastic gradient noise and curvature.

3. **Diffusion-modulation intensity**:

$$\boldsymbol{\lambda}^{(\ell)}(t) = \frac{\sqrt{\eta}}{\sigma_1^{(\ell)}(t)} \left[\boldsymbol{\Sigma}_t^{(\ell)}\right]^{1/2\top} \boldsymbol{J}_{op}^{(\ell)}(t), \tag{52}$$

which controls the variance of Lipschitz evolution by scaling the Wiener noise term from mini-batch sampling.

Luo et al. (2025a, Theorem 16) show that, at the network level, these quantities aggregate as:

$$\mu_Z(t) = \sum_{\ell=1}^{L} \mu^{(\ell)}(t), \qquad \kappa_Z(t) = \sum_{\ell=1}^{L} \kappa^{(\ell)}(t), \qquad \lambda_Z(t) = \left[\sum_{\ell=1}^{L} \|\boldsymbol{\lambda}^{(\ell)}(t)\|_2^2\right]^{\frac{1}{2}}, \tag{53}$$

governing the deterministic trend, irreversible growth, and stochastic fluctuations of the overall Lipschitz bound $K(t)$.

Their framework is particularly useful for interpreting the behaviors of neural networks, such as the near-convergence behavior, noisy supervision and mini-batch sampling trajectories. Luo et al. (2025a, § 8) show that:

1. The Lipschitz constant bound irreversibly increases since the term *noise-curvature entropy production* $\boldsymbol{\kappa}_Z(t)$ is *non-negative*, which increases the system entropy (Luo et al., 2025a, § 8.3).

2. The magnitude of uniform label noise affects the Lipschitz bound (Luo et al., 2025a, § 8.4 & 8.5). In particular, larger supervision noise leads to a lower Lipschitz bound for the network. This is because the supervision noise shrinks the *optimization-induced drift* $\boldsymbol{\mu}_Z(t)$.

3. Mini-batch trajectories do not affect the variance of the Lipschitz bound if batch size is sufficiently large (Luo et al., 2025a, § 8.6).

*Remark* 2.27. The dynamical analysis of Lipschitz continuity in deep learning models remains an open research problem. For instance, the dynamics under small-batch training remains poorly understood.

## 3 Estimation Methods

Proposition 2.19 gives an upper bound of the Lipschitz constant of a network (Miyato et al., 2018; Luo et al., 2025a). However, exact computation of the Lipschitz constant

$$K = \sup_{x \neq y} \frac{\|f(x) - f(y)\|_2}{\|x - y\|_2} = \sup_x \|\nabla f(x)\|_2$$

is generally NP-hard (Virmaux & Scaman, 2018), and even inapproximable under the Exponential Time Hypothesis (Jordan & Dimakis, 2020). This section surveys major estimation and bounding techniques, ranging from statistical sampling to certified convex relaxations. We group them into *Power Iteration*, *Extreme Value Theory*, *Derivative Bound Propagation*, *Spectral Alignment*, *Convex Optimization Relaxations*, and *Exact/Relaxed MILP formulations*.

### 3.1 Power Iteration for Single Linear Unit

**Power Iteration (Mises & Pollaczek-Geiringer, 1929)** is also known as the Von Mises iteration (Mises & Pollaczek-Geiringer, 1929). It is the standard way to estimate the Lipschitz constant for a linear layer — fully-connected or convolutional, by approximating the spectral-norm of weight matrices, which bounds the layer Lipschitz constant (Proposition 2.17) Yoshida & Miyato (2017); Miyato et al. (2018); Sedghi et al. (2019); Kim et al. (2021); Luo et al. (2025a). Another desirable property of *power iteration* in optimization is that *power iteration* is differentiable. For example, Yoshida & Miyato (2017) and Miyato et al. (2018) use *power iteration* for computing the largest singular values of parameter matrices, in which the differentiability of *power iteration* allows the gradient propagation for gradient based optimization.

Suppose that $W \in \mathbb{R}^{m \times m}$ is the parameter matrix of a fully-connected layer or a convolutional layer, then the largest singular value $\sigma_1(W)$ is its operator norm:

$$\text{Lip}[W] = \sigma_1(W). \tag{54}$$

Computing the exact value of $\sigma_1(W)$ for a large $m \times m$ matrix $W$ is computationally expensive, as classical algorithms require $O(m^3)$ time. However, the largest singular value can be approximated using *power iteration* by assuming the existence of a uniquely dominant singular value (Golub & Van Loan, 2013, § 7.3). Starting with random unit vectors $u_0, v_0$, one step of power iteration updates

$$v_{k+1} \;\leftarrow\; \frac{W^\top u_k}{\|W^\top u_k\|_2}, \qquad u_{k+1} \;\leftarrow\; \frac{W v_k}{\|W v_k\|_2}, \tag{55}$$

and uses

$$\hat{\sigma}_1^{(k+1)} = u_{k+1}^\top W v_{k+1} \tag{56}$$

as a differentiable estimate of $\sigma_1(W)$ at step $k + 1$. At the iteration number $T$, the estimated Lipschitz constant of $W$ is approximated as:

$$\text{Lip}[W] \approx u_T^\top W v_T, \tag{57}$$

where $T$ is the number of iterations. The estimation error is proportional to:

$$|\sigma_1 - \hat{\sigma}_1^{(T)}| = O\left(\left[\frac{\sigma_2}{\sigma_1}\right]^T\right), \tag{58}$$

where $\sigma_1$ and $\sigma_2$ are the largest, and second largest singular values, respectively (Golub & Van Loan, 2013, Equation 7.3.5, § 7.3).

### 3.2 Extreme Value Theory

**CLEVER (Weng et al., 2018b)** — *Cross-Lipschitz Extreme Value for nEtwork Robustness* — converts the problem of estimating the attack-independent robustness into the problem of estimating the *local* Lipschitz constant by sampling gradient norms in a neighborhood of the input (Weng et al., 2018b). This is because the $q$-norm Lipschitz constant of a network $g : \mathbb{R}^m \to \mathbb{R}$ is determined by:

$$\text{Lip}_q[g] = \sup_x \|\nabla g(x)\|_q, \tag{59}$$

so that, for any inputs $x$ and $y$, the inequality holds:

$$|g(x) - g(y)| \leq \text{Lip}_q[f]\, \|x - y\|_p,$$

(Weng et al., 2018b, Lemma 3.1) where $1 \leq p, q \leq \infty$ satisfy the Hölder conjugacy relation (Rudin, 1976, § 6):

$$\frac{1}{p} + \frac{1}{q} = 1.$$

**Adversarial Perturbation Bound.** Weng et al. (2018b) further show that the local Lipschitz constant can be used for deriving the adversarial perturbation bound. Let

$$f : \mathbb{R}^m \to \mathbb{R}^k$$

be a $k$-class classifier and $f_i$ be the $i$-th prediction. Let:

$$c(x) = \arg\max_{1 \leq i \leq k} f_i(x)$$

be the prediction. Then the $p$-norm adversarial perturbation $\|\delta\|_p$ to an input $x_0$ is bounded above by:

$$\|\delta\|_p \leq \min_{j \neq c} \frac{f_c(x_0) - f_j(x_0)}{\mathrm{Lip}_q [f_c - f_j]},$$

where the local Lipschitz constant $\mathrm{Lip}_q [f_c - f_j]$ can be estimated through:

$$\mathrm{Lip}_q [f_c - f_j] = \max_{x \in B_p(x_0, \delta)} \|\nabla (f_c - f_j)(x)\|_q \tag{60}$$

and $B_p(x_0, \delta)$ is a $p$-norm ball centered at $x_0$ with a radius $\delta$ (Weng et al., 2018b, Theorem 3.2).

**Distribution of Extreme Value** $\max_{x \in B_p(x_0, \delta)} \|\nabla (f_c - f_j)(x)\|_q$. Estimating:

$$\max_{x \in B_p(x_0, \delta)} \|\nabla_x (f_c - f_j)(x)\|_q \tag{61}$$

can be through sampling $x \in B_p(x_0, \delta)$. Suppose the $n$ samples are $\{x^{(1)}, x^{(2)}, \cdots, x^{(n)}\}$. Their gradient $q$-norms are $\{\|\nabla (f_c - f_j)(x^{(1)})\|_q, \|\nabla (f_c - f_j)(x^{(2)})\|_q, \cdots, \|\nabla (f_c - f_j)(x^{(n)})\|_q\}$. Weng et al. (2018b) show that the extreme value of the gradient $q$-norm:

$$Y := \lim_{n \to \infty} \max_{x^{(i)} \in B_p(x_0, \delta)} \left\{ \|\nabla (f_c - f_j)(x^{(1)})\|_q, \|\nabla (f_c - f_j)(x^{(2)})\|_q, \cdots, \|\nabla (f_c - f_j)(x^{(n)})\|_q \right\} \tag{62}$$

can only be a distribution $P_Y$ of three distribution classes:

- **Type I**: *Gumbel class*,

- **Type II**: *Fréchet class*,

- **Type III**: *Reverse Weibull class*,

according to Fisher-Tippett-Gnedenko Theorem (Weng et al., 2018b, Theorem 4.1). The extreme value $\max \|\nabla (f_c - f_j)(x^{(1)})\|_q$ is thus converted to estimating the right end-point (the location parameter $a_W$) of the reverse Weibull distribution $P_Y$:

$$\mathrm{Lip}_q [f_c - f_j] \approx a_W. \tag{63}$$

### 3.3 Coordinate-Wise Gradient

**Fast-Lip (Weng et al., 2018a)** derives an upper bound of local Lipschitz constant in the form of coordinate-wise gradients by analyzing the activation patterns of ReLU networks. Let $n_k$ denote the number of neurons at the $k$-th layer of an $m$-layer network and Let $n_0$ be the input dimension. Let $\phi_k : \mathbb{R}^{n_0} \to \mathbb{R}^{n_k}$ be the map from input to the output of $k$-th layer. Let $\rho$ be the coordinate-wise activation function. Then the relation between the $(k-1)$-th layer and the $k$-th layer can be written as:

$$\phi_k(x) = \rho\Big(W^{(k)}\phi_{k-1}(x) + b^{(k)}\Big), \tag{64}$$

where $W^{(k)} \in \mathbb{R}^{n_k \times n_{k-1}}$ is the $k$-th layer parameter matrix, and $b^{(k)} \in \mathbb{R}^{n_k}$ is the bias. Set $f(x) = \phi_m(x)$ and let $f_j(x)$ denote the $j$-th output of $f$. Weng et al. (2018a) start from analyzing the ReLU activation patterns and then deriving gradient bounds under this setting for bounding local Lipschitz constant.

**Activation Patterns of ReLU Networks.** Let $l_r^{(k)}$ and $u_r^{(k)}$ denote the lower and upper bound for the $r$-th neuron in the $k$-th layer and let $z_r^{(k)}$ be the pre-activation at the $k$-th layer, given as:

$$z_r^{(k)} = W_{r,:}^{(k)}\phi_{k-1}(x) + b_r^{(k)}, \tag{65}$$

where $W_{r,:}^{(k)}$ denotes the $r$-th row of $W^{(k)}$ (Weng et al., 2018a, § 3.2). For the neurons indexed by $[n_k] := \{1, 2, \cdots, n_k\}$, then there are only three activation patterns:

1. **Always Activated.** Neurons are always activated: $\mathcal{I}_k^+ := \{r \in [n_k] \mid u_r^{(k)} \geq l_r^{(k)} \geq 0\}$.

2. **Always Inactivated.** Neurons are always inactivated: $\mathcal{I}_k^- := \{r \in [n_k] \mid l_r^{(k)} \leq u_r^{(k)} \leq 0\}$.

3. **Either Activated or Inactivated.** Neurons are either activated or inactivated: $\mathcal{I}_k := \{r \in [n_k] \mid l_r^{(k)} \leq 0 \leq u_r^{(k)}\}$.

**Gradient Analysis of ReLU Networks.** Gradient norm carries the information for local Lipschitz constant. Weng et al. (2018a) then analyze the coordinate-wise gradients of ReLU networks with a manner of layer-wise. Using the activation patterns of ReLU networks, the $k$-th layer output can be rewritten into:

$$\phi_k(x) = \Lambda^{(k)}\Big(W^{(k)}\phi_{k-1}(x) + b^{(k)}\Big), \tag{66}$$

where $\Lambda^{(k)}$ is the activation pattern matrix:

$$\Lambda_{r,r}^{(k)} = \begin{cases} 1 \text{ or } 0, & \text{if } r \in \mathcal{I}_k \\ 1, & \text{if } r \in \mathcal{I}_k^+ \\ 0, & \text{if } r \in \mathcal{I}_k^- \end{cases}. \tag{67}$$

Let $\Lambda_a^{(k)}$ be the diagonal activation matrix for neurons in the $k$-th layer that are always activated and set $\Lambda_u^{(k)} := \Lambda^{(k)} - \Lambda_a^{(k)}$. Starting by analyzing the bound of the gradient $\nabla\phi_k(x)$ coordinate-wisely in a 2-layer ReLU network, Weng et al. (2018a) show that an inequality for gradient $q$-norm holds:

$$\max_{x \in B_p(x_0, \epsilon)} \left|[\nabla f_j(x)]_k\right| \leq \max\Big(C_{j,k}^{(1)} + L_{j,k}^{(1)}, \; C_{j,k}^{(1)} + U_{j,k}^{(1)}\Big), \tag{68}$$

where:

$$C_{j,k}^{(1)} = W_{j,:}^{(2)}\Lambda_a^{(1)}W_{:,k}^{(1)}, \quad L_{j,k}^{(1)} = \sum_{i \in \mathcal{I}_1, W_{j,i}^{(2)}W_{i,k}^{(2)} < 0} W_{j,i}^{(2)}W_{i,k}^{(2)}, \quad \text{and} \quad U_{j,k}^{(1)} = \sum_{i \in \mathcal{I}_1, W_{j,i}^{(2)}W_{i,k}^{(2)} > 0} W_{j,i}^{(2)}W_{i,k}^{(2)}. \tag{69}$$

Then the gradient $q$-norm can be bounded above by:

$$\text{Lip}_q\left[f_j; B_p(x_0, \epsilon)\right] = \max_{x \in B_p(x_0, \epsilon)} \|\nabla f_j(x)\|_q \leq \left[\sum_k \Big(\max_{x \in B_p(x_0, \epsilon)} \left|[\nabla f_j(x)]_k\right|\Big)^q\right]^{\frac{1}{q}}, \tag{70}$$

yielding a local Lipschitz constant upper bound. This method is referred to as **Fast-Lip** in Weng et al. (2018a).

### 3.4 Spectral Alignment

**SeqLip (Virmaux & Scaman, 2018)** refines the upper bound of an $L$-layer sequential network $f \in \mathbb{R}^{n_1} \to \mathbb{R}^{n_L}$ with 1-Lipschitz activation functions:

$$\text{Lip}\,[f]^+ = \prod_{\ell=1}^{L} \|W^{(\ell)}\|_2, \tag{71}$$

where $W^{(\ell)} \in \mathbb{R}^{n_\ell \times n_{\ell+1}}$ is the $\ell$-th layer parameter matrix, and $\text{Lip}\,[f]^+$ is referred to as **AutoLip** bound in Virmaux & Scaman (2018). By taking into account the spectral alignment between the parameter matrices of two consecutive layers, the bound is then refined as:

$$\text{Lip}\,[f] \le \text{Lip}\,[f]^+ \underbrace{\left( \prod_{\ell}^{L-1} \sqrt{(1 - r_\ell - r_{\ell+1}) \max_{t^{(\ell)} \in [0,1]^{n_\ell}} \langle t^{(\ell)} \cdot v_1^{(\ell+1)}, \, u_1^{(\ell)} \rangle^2 + r_\ell + r_{\ell+1} + r_\ell r_{\ell+1}} \right)}_{\text{spectral alignment}}, \tag{72}$$

where $t^{(\ell)} \in [0,1]^{n_\ell}$ represents the $\ell$-th layer gradient patterns from activation functions for $n_\ell$ neurons — the coordinate-wise gradient of an activation function such as ReLU falls in $[0,1]$, $u_1^{(\ell)}$ is the first left-singular vector for $W^{(\ell)}$, $v_1^{(\ell+1)}$ is the first right-singular vector for $W^{(\ell+1)}$, and $r_\ell = \frac{\sigma_2^{(\ell)}}{\sigma_1^{(\ell)}}$ is the ratio of the second largest singular value to the first largest singular value for $W^{(\ell)}$ (Virmaux & Scaman, 2018, Theorem 3).

Virmaux & Scaman (2018) also show that, if $u, v \in \mathbb{R}^n$ are two independent random vectors taken uniformly from $\mathbb{S}^{n-1} = \{x \in \mathbb{R}^n \mid \|x\|_2 = 1\}$, the following limit:

$$\lim_{n \to \infty} \max_{t \in [0,1]^n} \left| \langle t \cdot u, v \rangle \right| = \frac{1}{\pi} \tag{73}$$

holds *almost surely*. Under this independent assumption, the bound reduces to:

$$\text{Lip}\,[f] \approx \frac{\text{Lip}\,[f]^+}{\pi^{L-1}} \tag{74}$$

(Virmaux & Scaman, 2018, Lemma 2). In real setting, the singular vectors are not independent across layers.

### 3.5 Convex Optimization Relaxation

**LipSDP (Fazlyab et al., 2019)** interprets activation functions as gradients of convex potential functions, satisfying certain properties described by quadratic constraints. Therefore, Lipschitz constant estimation problem is treated as a semi-definite program (SDP) problem.

Let $f : \mathbb{R}^{n_0} \to \mathbb{R}^{n_L}$ be an $L$-layer feed-forward network, recursively defined as:

$$\begin{cases} x^{(0)} = x \\ x^{(k+1)} = \rho\left(W^{(k)} x^{(k)} + b^{(k)}\right) \end{cases}, \tag{75}$$

where $x^{(\ell)}$ is the output at the $\ell$-th layer, $W^{(\ell)} \in \mathbb{R}^{n_{\ell+1} \times n_\ell}$ is the parameter matrix at the $\ell$-th layer, $b^{(\ell)} \in \mathbb{R}^{n_{\ell+1}}$ is the bias at the $\ell$-th layer, and $\rho$ is the coordinate-wise activation function. For a single layer network:

$$f(x) = W^{(1)} \rho\left(W^{(0)} x + b^{(0)} + b^{(1)}\right), \tag{76}$$

Fazlyab et al. (2019) shows that the Lipschitz constant of $f$ is given by a SDP problem:

$$\text{Lip}\,[f] \le \sqrt{t}, \tag{77}$$

defined by:

$$\text{minimize} \quad t \tag{78}$$

$$\text{subject to} \quad M(t,T) \preceq 0 \quad T \in T_n, \tag{79}$$

where $\rho$ is *slope-restricted* on $[\alpha, \beta]$ (Fazlyab et al., 2019, Definition 1):

$$\alpha \leq \frac{\rho_j(x) - \rho_j(y)}{x - y} \leq \beta \qquad \forall x, y \in \mathbb{R}, \tag{80}$$

$T_n$ is a convex set (Fazlyab et al., 2019, Lemma 1):

$$T_n := \Big\{ T \in \mathbb{S}^n \mid T = \sum_{i=1}^n \lambda_{ii} e_i e_i^\top, \lambda_{ii} \geq 0 \Big\}, \tag{81}$$

and:

$$M(t,T) := \begin{bmatrix} -2\alpha\beta(W^{(0)})^\top T W^{(0)} - t I_{n_0} & (\alpha+\beta)(W^{(0)})^\top T \\ (\alpha+\beta)T W^{(0)} & -2T + (W^{(1)})^\top W^{(1)} \end{bmatrix} \preceq 0 \tag{82}$$

holds true for $T \in T_n$ (Fazlyab et al., 2019, Theorem 1). For more general $L$-layer fully-connected network, Fazlyab et al. (2019) show that $M(t,T)$ is:

$$M(t,T) = \begin{bmatrix} A \\ B \end{bmatrix}^\top \begin{bmatrix} -2\alpha\beta T & (\alpha+\beta)T \\ (\alpha+\beta)T & -2T \end{bmatrix} \begin{bmatrix} A \\ B \end{bmatrix} + \begin{bmatrix} -t I_{n_0} & 0 & \cdots & 0 \\ 0 & 0 & \cdots & 0 \\ \vdots & \vdots & \ddots & \vdots \\ 0 & 0 & \cdots & (W^{(L)})^\top W^{(L)} \end{bmatrix} \preceq 0, \tag{83}$$

where:

$$A = \begin{bmatrix} W^{(0)} & 0 & \cdots & 0 & 0 \\ 0 & W^{(1)} & \cdots & 0 & 0 \\ \vdots & \vdots & \ddots & \vdots & \vdots \\ 0 & 0 & \cdots & W^{(L-1)} & 0 \end{bmatrix}, \qquad B = \begin{bmatrix} 0 & I_{n_1} & 0 & \cdots & 0 \\ 0 & 0 & I_{n_2} & \cdots & 0 \\ \vdots & \vdots & \vdots & \ddots & \vdots \\ 0 & 0 & 0 & \cdots & I_{n_L} \end{bmatrix}, \tag{84}$$

and:

$$C = \begin{bmatrix} 0 & \cdots & 0 & W^{(L)} \end{bmatrix}, \qquad b = \begin{bmatrix} (b^{(0)})^\top \cdots (b^{(L-1)})^\top \end{bmatrix}^\top \tag{85}$$

(Fazlyab et al., 2019, Theorem 2).

### 3.6 Integer Programming for ReLU Networks

It is well known that computing the Lipschitz constant of an arbitrary scalar- or vector-valued function is NP-hard (Virmaux & Scaman, 2018), and even inapproximable under the Exponential Time Hypothesis (Jordan & Dimakis, 2020). A common approximation is to estimate the Lipschitz constant using gradient norms, where the Jacobians can be explicitly derived via the chain rule. However, nondifferentiabilities, *e.g.*, those arising in ReLU networks, can introduce inaccuracies into these estimates. An alternative approach is to formulate the Lipschitz bounding problem as an integer programming problem, which avoids such inaccuracies by directly bounding the Jacobians.

**LipMIP (Jordan & Dimakis, 2020)** is a method that provably exactly compute $\ell_1$- and $\ell_\infty$-norm Lipschitz constants of non-smooth ReLU networks. We summarize their method by simplifying their notations. Let $f : \mathbb{R}^d \to \mathbb{R}$ be a ReLU network. Let $\rho$ be activation function. Then, an $L$-layer ReLU network is recursively defined as:

$$\begin{cases} f(x) = \rho(Z_L(x)) \\ Z^{(\ell)}(x) = W^{(\ell)} \rho\Big( Z^{(\ell-1)}(x) \Big) + b^{(\ell)} \\ Z^{(0)} = x \end{cases} , \tag{86}$$

where $W^{(\ell)} \in \mathbb{R}^{n_\ell \times n_{\ell-1}}$ is the $\ell$-th layer parameter matrix, and $Z^{(\ell)}(x)$ is the output of the $\ell$-th layer neurons. Let $\partial f(x)$ be the Clarke sub-differential convex hull of $f$ at point $x$ — see Section 2.4 (Sub-Differential Convex Functions). For example:

$$\partial \rho(0) = [0, 1]. \tag{87}$$

**Pushforward Sub-Differential Hull Set.** However, in practice, PyTorch implementations often set $\partial \rho(0) = 1$, which can lead to inaccuracies in estimating the Lipschitz constant. To correctly estimate the Lipschitz constant of a ReLU network, the analysis must be carried out in the sense of sub-differentials. Let $\partial f(X)$:

$$\partial f(X) = \left\{ g \mid g \in \partial f(x), x \in X \right\} \tag{88}$$

be the set-valued Clarke sub-differentials at set $X$. Let $\nabla^\sharp f(\bullet)$ be the pushforward set that the sub-gradients of ReLU activation at value 0 come from the hull set $\partial \rho(0)$:

$$\nabla^\sharp f(\bullet) = \left\{ \nabla f(\bullet) \mid \text{the sub-differentials of activations come from the hull set } \partial \rho(0) \right\}. \tag{89}$$

Thus for a ReLU network, the *feasible set* for its gradients on $X$ is given as:

$$\nabla^\sharp f(X) = \left\{ G \in \nabla^\sharp f(x) \mid x \in X \right\}. \tag{90}$$

Theoretically, (Jordan & Dimakis, 2020) show that the push forward set sub-gradient $\nabla^\sharp f(x)$ is the Clarke sub-gradient hull $\partial f(x)$, that is:

$$\nabla^\sharp f(x) = \partial f(x) \tag{91}$$

(Jordan & Dimakis, 2020, Theorem 2) with all gradients come from $\partial \rho(0)$. Then the local $p \to q$ Lipschitz constant of $f$ on $X$ is given as:

$$\mathrm{Lip}_{p \to q}[f; X] = \sup_{G \in \nabla^\sharp f(X)} \sup_{\|v\|_p \le 1} \frac{\|G^\top v\|_q}{\|v\|_p} = \sup_{G \in \nabla^\sharp f(X)} \|G^\top\|_q. \tag{92}$$

Finding the Lipschitz constant on $X$ can be formulated as an integer programming problem on the feasible set $\nabla^\sharp f(X)$.

**Solving the Supremum of Sub-Differential Hull Set.** The problem:

$$\sup_{G \in \nabla^\sharp f(X)} \left\{ \|G^\top\|_q \mid G \in \nabla^\sharp f(X) \right\} \tag{93}$$

is then represented as a *mixed-integer polytope*, which is a mixed-integer programming problem for maximizing $\|G^\top\|_q$ by solving the combinations of input $x \in X$ and ReLU's sub-gradient $a \in [0,1]^n$ in $X \times [0,1]^n$ (Jordan & Dimakis, 2020, Definition 6 & Lemma 1). This problem can be solved by off-the-shelf MIP solvers.

## 3.7 Remarks

The estimation methods surveyed above involve an inherent trade-off between computational efficiency, tightness, and scalability. Power iteration (Mises & Pollaczek-Geiringer, 1929; Yoshida & Miyato, 2017; Miyato et al., 2018) (see Section 3.1) is lightweight and differentiable, but it only provides a layer-wise upper-bound estimate of the spectral norm, and therefore may induce a loose network-level upper bound when combined across layers (see Proposition 2.17 and Proposition 2.19). Extreme-value-theoretic approaches such as CLEVER (Weng et al., 2018b) (see Section 3.2) are scalable and often yield tight empirical estimates of local Lipschitz behavior by leveraging the extreme value distributions of observed Lipschitz constants (see Lemma 2.5), yet they do not provide formal certificates. Methods based on coordinate-wise gradient

propagation and spectral alignment, such as Fast-Lip (Weng et al., 2018a) (see Section 3.3) and SeqLip (Virmaux & Scaman, 2018) (see Section 3.4), improve upon spectral-norm product bounds by exploiting architectural structure (see Proposition 2.19), but they remain approximate and are typically tailored to specific network classes. In contrast, convex relaxation methods such as LipSDP (Fazlyab et al., 2019) (see Section 3.5) and mixed-integer formulations such as LipMIP (Jordan & Dimakis, 2020) (see Section 3.6) offer substantially stronger guarantees, with the latter being exact for ReLU networks on bounded domains by working directly with the feasible Jacobian or sub-differential set (see Section 2.4); however, these methods are considerably more computationally expensive and generally require additional architectural information to remain tractable.

## 4 Regularization Approaches

A variety of techniques have been developed to explicitly or implicitly enforce Lipschitz continuity in deep neural networks. These approaches can be categorized into:

(i) Section 4.1: **Weight Regularization** — constraining or regularizing weight matrices during initialization and training;

(ii) Section 4.2: **Gradient Regularization** — penalizing or normalizing gradient norms during training;

(iii) Section 4.3: **Activation Regularization** — designing or constraining activation functions to be Lipschitz-bounded;

(iv) Section 4.4: **Class-Margin Regularization** — implicitly enforcing Lipschitz constraints via maximizing class decision boundaries;

(v) Section 4.5: **Architectural Regularization** — enforcing Lipschitz constraints by designing architectures, particularly for transformers.

### 4.1 Weight Regularization

This subsection reviews methods that control the Lipschitz constant by directly constraining or regularizing the network's weight parameters, either at initialization or during training.

#### 4.1.1 Weight Clipping

Weight clipping is a straightforward method to enforce Lipschitz continuity by limiting the magnitude of weight matrices. This is because for a matrix $W$, the operator-norm variation $\|W\|_{op}$ in $\ell_2$ for $W$ under perturbation $\Delta W$ is bounded above by:

$$\|W\|_{op} = \sigma_1(W) = \langle u_1 v_1^\top, \, \Delta W \rangle \leq \|u_1 v_1^\top\|_2 \, \|\Delta W\|_2, \tag{94}$$

with Cauchy–Schwarz inequality, where $\sigma_1(W)$ is the largest singular value of $W$, $u_1, v_1$ are the left- and right-singular vectors corresponding to the largest singular value (Luo et al., 2025b, Lemma 5.1).

In the setting of generative adversarial networks (GANs), Arjovsky et al. (2017) showthat the discriminator $f$ of a GAN evaluates the Wasserstein distance $W(\mathbb{P}, \mathbb{Q})$ between two distributions $\mathbb{P}$ and $\mathbb{Q}$. The Kantorovich-Rubinstein duality (Villani et al., 2008) tells that:

$$W(\mathbb{P}, \mathbb{Q}) = \frac{1}{K} \sup_{\mathrm{Lip}[f] \leq K} \mathbb{E}_{x \sim \mathbb{P}}\big[f(x)\big] - \mathbb{E}_{y \sim \mathbb{Q}}\big[f(y)\big] \tag{95}$$

(Arjovsky et al., 2017). Therefore reducing the Lipschitz constant $K$ can reduce the Wasserstein distance $W(\mathbb{P}, \mathbb{Q})$. Arjovsky et al. (2017) propose clipping weights to a fixed range:

$$W \leftarrow \mathrm{clip}(W, -c, +c) \tag{96}$$

(Arjovsky et al., 2017, Algorithm 1), where $c$ is a small positive constant (*e.g.*, $c = 0.01$). Weight clipping is computationally efficient, but can lead to exploding or vanishing gradients if $c$ is too small, reducing model capacity, as criticized in later literature (Gulrajani et al., 2017).

### 4.1.2 Spectral Normalization and Regularization

**Explicit Regularization by Penalizing Spectral Norm.** Referring to Propositions 2.17 (Lipschitz Constant of a Linear Layer) and 2.19 (Lipschitz Constant Upper Bound of Feedforward Neural Network), the product of the spectral norms of the parameter matrices gives rise to an upper bound on the Lipschitz constant of a neural network. To penalize this Lipschitz bound during training and thereby improve generalization, Yoshida & Miyato propose, for an $L$-layer network parameterized by:

$$W := (W^{(1)}, W^{(2)}, \cdots, W^{(L)}),$$

the following regularization objective:

$$\min_{W} \mathscr{L}_{\text{task}}(\xi; W) + \frac{\lambda}{2} \sum_{\ell=1}^{L} \left(\sigma_1^{(\ell)}\right)^2, \tag{97}$$

where $\mathscr{L}_{\text{task}}(\xi; W)$ represents the task loss evaluated on mini batch $\xi$, $\lambda$ denotes regularization coefficient and $\sigma_1^{(\ell)}$ denotes the largest singular value of the $\ell$-th parameter matrix (Yoshida & Miyato, 2017, Equation 1). In implementation, each $\sigma_1^{(\ell)}$ is estimated using power iteration (see Section 3.1).

**Implicit Regularization by Normalizing Spectral Norm.** A challenge for Generative Adversarial Networks (GANs) (Goodfellow et al., 2014) is that the prediction of discriminator is often inaccurate and unstable during training (Arjovsky & Bottou, 2017; Miyato et al., 2018). Miyato et al. (2018) propose normalizing the weight matrix $W$ of a linear layer by:

$$W \leftarrow \frac{W}{\|W\|_2} = \frac{W}{\sigma_1}, \tag{98}$$

where $\sigma_1$ is the largest singular value of parameter matrix $W$, for ensuring a Lipschitz constant of at most 1 for the layer (Miyato et al., 2018, Equation 8). Readers can also refer to Proposition 2.18. This method was proposed by Miyato et al. for training generative adversarial networks (GANs) (Goodfellow et al., 2014), improving the generations (Miyato et al., 2018, § 2.1).

### 4.1.3 Orthogonal Weight

**Parseval Networks (Cisse et al., 2017).** Parseval networks enforce approximate orthonormality on weight matrices to bound their spectral-norms (Cisse et al., 2017). For a weight matrix $W^{(\ell)}$ in layer $\ell$, Cisse et al. (2017) minimize the 2-norm of the deviation from orthonormality:

$$\|W^{(\ell)\top} W^{(\ell)} - I\|_F, \tag{99}$$

ensuring $\|W^{(\ell)}\|_2 \leq 1$ (Cisse et al., 2017, § 4.2). For a feedforward network with $L$ layers and 1-Lipschitz activations (*e.g.*, ReLU), the global Lipschitz constant is bounded by:

$$K \leq \prod_{\ell=1}^{L} \|W^{(\ell)}\|_2 \leq 1, \tag{100}$$

(see Proposition 2.19 (Lipschitz Constant Upper Bound of Feedforward Neural Network)). (Cisse et al., 2017) argue that the robustness to adversarial attacks is enhanced by the reduced sensitivity to perturbations.

## 4.2 Gradient Regularization

Gradient-based regularization methods enforce Lipschitz continuity by constraining the gradient norm of the loss function or network output with respect to inputs, ensuring smooth decision boundaries and robustness. Note that, for a network $f$ on convex $X$, its 2-norm Lipschitz constant on $X$ is given by:

$$\text{Lip}\,[f] = \sup_{x \in X} \|\nabla f(x)\|_2, \tag{101}$$

(see Lemma 2.5 (Lipschitz Constant Bounds Gradient Norm)). Theoretically, constraining $\|\nabla f(x)\|_2$ can bound Lipschitz constant of $f$.

For example, Gulrajani et al. (2017) argue that bounding Lipschitz constant by weight clipping can lead to exploding or vanishing gradients, and reduce capacity of the critic in a GAN. They then introduce a regularization method by directly bounding the gradients of the critic network $D$ through:

$$\mathbb{E}_{x \sim \mathbb{P}_X}\left[\left(\|\nabla D(x)\|_2 - 1\right)^2\right], \tag{102}$$

where $\mathbb{P}_X$ is the distribution of training sample set $X$ (Gulrajani et al., 2017, § 4).

## 4.3 Activation Regularization

Activation regularization methods modulate Lipschitz continuity by constraining the properties of activation functions or their outputs. Such approaches may impose explicit norm bounds, design activations that are inherently norm-preserving, or apply penalties to activation magnitudes, thereby limiting the contribution of nonlinearities to the overall Lipschitz constant of the network.

### 4.3.1 Group-Sorting Activation

Bounding the Lipschitz constant of a network by ensuring 1-Lipschitz for affine units through spectral-norm constraints can improve robustness (Yoshida & Miyato, 2017; Miyato et al., 2018). However, this often comes at the cost of reduced expressiveness (Anil et al., 2019). Anil et al. (2019) first show that, to remain expressive in a spectral-norm constrained network, the network must preserve the gradient norms at each layer (Anil et al., 2019). ReLU networks can only satisfy this for positive values, this is because:

$$\frac{\partial \text{ReLU}(x)}{\partial x} = \begin{cases} 1, & x > 0 \\ (0,1), & \text{in sub-differential sense .} \\ 0, & x < 0 \end{cases} \tag{103}$$

To preserve the gradient norms at activation layer, Anil et al. (2019) use a general purpose 1-Lipschitz activation function **GroupSort**$(x)$ which is homogeneous:

$$\textbf{GroupSort}(\alpha\,x) = \alpha\,\textbf{GroupSort}(x) \tag{104}$$

(Anil et al., 2019). **GroupSort** activation separates a pre-activation $x$ into $k$ groups:

$$x = (\underbrace{x_1, \cdots, x_{g_1}}_{\text{group 1}}, \underbrace{x_{g_1+1}, \cdots, x_{g_2}}_{\text{group 2}}, \cdots \underbrace{x_{g_{k-1}+1}, \cdots, x_{g_k}}_{\text{group k}}), \tag{105}$$

and sorts each group into ascending order:

$$\textbf{GroupSort}(x) = (\underbrace{x_{s_1}, \cdots, x_{s_{g_1}}}_{\text{group 1}}, \underbrace{x_{s_{g_1}+1}, \cdots, x_{s_{g_2}}}_{\text{group 2}}, \cdots \underbrace{x_{s_{g_{k-1}}+1}, \cdots, x_{s_{g_k}}}_{\text{group k}}), \tag{106}$$

where the $i$-th sorted group satisfies:

$$x_{s_{g_{i-1}}+1} \leq x_{s_{g_{i-1}}+2} \leq \cdots \leq x_{s_{g_i}} \tag{107}$$

(Anil et al., 2019). For the pre-activation in a linear unit:

$$Wx \tag{108}$$

where $W \in \mathbb{R}^{m \times n}$ and $x \in \mathbb{R}^n$, if the linear unit is with spectral-norm constraint such that:

$$\|W\|_2 = 1, \tag{109}$$

**GroupSort** activation directly gives rise to a 1-Lipschitz constant activation:

$$\sup_{\|x\|_2=1} \|\textbf{GroupSort}(Wx)\|_2 = \sup_{\|x\|_2=1} \|Wx\|_2 = \|W\|_2. \tag{110}$$

When the group size is 2, Anil et al. (2019) refers to this special case as **MaxMin**, which is equivalent to the Orthogonal Permutation Linear Unit (OPLU) (Chernodub & Nowicki, 2017); when the group size is the entire input, this is referred to as **FullSort** in the literature (Chernodub & Nowicki, 2017).

To train a network with **GroupSort** activations and guarantee that all linear units are exactly 1-Lipschitz during optimization instead of bounded by 1 (Cisse et al., 2017; Gulrajani et al., 2017), Chernodub & Nowicki (2017) approximate the updated parameter matrix $W_0$ with a closest orthogonal matrix through a differentiable, iterative algorithm, referred to as *Björck Orthogonalization* (Björck & Bowie, 1971). This algorithm is defined by:

$$W_{k+1} = W_k \left( I + \frac{1}{2} Q_k + \cdots + (-1)^p \binom{-\frac{1}{2}}{p} Q_k^p \right), \tag{111}$$

where $W_k$ is the $k$-th iterative result, $p$ is a chosen hyper-parameter and $Q_k$ is:

$$Q_k = I - W_k^\top W_k \tag{112}$$

(Chernodub & Nowicki, 2017, § 4.2.1). As a result, the iteration at step $T$ leads to:

$$W_T^\top W_T \approx I, \tag{113}$$

which provably ensures the linear unit with parameter matrix $W$ is 1-Lipschitz continuous:

$$\|W_T\|_2 \approx 1. \tag{114}$$

### 4.3.2   Contraction Activation and Invertible Residual Map

Flow-based models or flows learn to transform a source distribution $p_X$ to target distribution $p_Z$, consisting of invertible networks (Rezende & Mohamed, 2015). Ensuring that the transformation is invertible and its Jacobian is computable is a straightforward way for ensuring invertibility:

$$\log p_X(x) = \log p_Z(z) + \log |\det J_F(x)| \tag{115}$$

where $F : \mathbb{R}^m \to \mathbb{R}^m$ is an invertible map and $J_F(x)$ is the Jacobian of $F$ at $x$ (Berg et al., 2018; Dinh et al., 2017). Different from explicit computation of Jacobian, Behrmann et al. (2019) propose a method for inverting residual networks (He et al., 2016) by ensuring the Lipschitz constants be less than 1, so that the residual networks are *contraction maps* (Behrmann et al., 2019; Perugachi-Diaz et al., 2021).

**Inverting Residual Layer.**   Let

$$F(x) = x + g(x) \tag{116}$$

be a residual layer where $F : \mathbb{R}^m \to \mathbb{R}^m$, $g : \mathbb{R}^m \to \mathbb{R}^m$ and $x \in \mathbb{R}^m$. Let $g$ be a contraction map so that:

$$\text{Lip}\,[g] < 1. \tag{117}$$

Suppose that the inversion $F^{-1}$ exists and set $y = F(x)$, so that:

$$x = F^{-1}(y) = y - g(x). \tag{118}$$

A **sufficient condition** for inverting $F(x)$ is that $g$ is a contraction map:

$$\text{Lip}[g] < 1 \tag{119}$$

(Behrmann et al., 2019, Theorem 1). Set:

$$T(x) = F^{-1}(y) = y - g(x), \tag{120}$$

then the inversion of $y$ is a fixed-point problem for solving:

$$x = T(x). \tag{121}$$

Using *Banach contraction principle* or *Banach fixed-point theorem*, set $x_0 = y$, the $F^{-1}(y)$ can be approximated through:

$$x_{k+1} = y - g(x_k) \tag{122}$$

iteratively.

**Contraction Activation.** Chen et al. (2019) first introduce **LipSwish** activation function for inverting residual networks, defined as:

$$\textbf{LipSwish}(x) = \frac{x\sigma(\beta\,x)}{1.1}, \tag{123}$$

where $\sigma$ is the sigmoid function, and $\beta > 0$ is a learnable positive constant (kept positive via softplus), initialized with 0.5. **LipSwish** has a Lipschitz constant at most 1. However, the negative axis of **LipSwish** has zero gradients almost everywhere. In a 1-Lipschitz continuous network, the Jacobian norm across two consecutive layers is reduced to be at most one, which limits the network's expressiveness and referred to as *gradient norm attenuation* problem (Anil et al., 2019; Li et al., 2019). To mitigate this problem, Perugachi-Diaz et al. (2021) use specially designed activation function **CLipSwish** by concatenating two **LipSwish** functions. The **CLipSwish** is therefore given as:

$$\Phi(x) = \begin{bmatrix} \textbf{LipSwish}(x) \\ \textbf{LipSwish}(-x) \end{bmatrix}, \qquad \textbf{CLipSwish}(x) = \frac{\Phi(x)}{\text{Lip}[\Phi]} \le 1, \tag{124}$$

where $\text{Lip}[\Phi] \approx 1.004$ (Perugachi-Diaz et al., 2021, § 3.4). This concatenation overcomes the gradient attenuation problem introduced by **LipSwish** since the negative axis has zero gradients almost everywhere while ensuring the activation is a contraction map.

## 4.4 Class-Margin Regularization

**Input Margin.** The goal of adversarial defense in classification task is to ensure that, for a $k$-class classifier $f : \mathbb{R}^d \to \mathbb{R}^k$, and an input $x \in \mathbb{R}^d$, a bounded perturbation $\|\delta_x\|_2 < c$ does not alter the prediction:

$$\hat{y} = \underset{i \in [k] = \{1, 2, \cdots, k\}}{\arg\max} f_i(x + \delta_x), \tag{125}$$

where $f_j(x)$ denotes the output at the $j$-th coordinate, and the $\delta_x$ giving rise to the minimal $\|\delta_x\|_2$ is often referred to as 2-norm *input margin* (Ngnawé et al., 2024). More generally, *input margin* can also be discussed with respect to $p$-norm ($1 \le p \le \infty$). The $p$-norm *input margin* represents the decision boundary in the input space with respect to the topology induced by $p$-norm.

**Class/Logit Margin.** Of the same setting, the *class margin* (*i.e. logit margin*) $m(x)$ is defined as:

$$m(x) = f_i(x) - \max_{j \neq i} f_j(x), \tag{126}$$

where $i$ is the ground-truth (Tsuzuku et al., 2018, § 4.1). By writing equation (equation 126) into:

$$m(x) = (e_i - e_j)^\top \begin{bmatrix} f_i(x) \\ \max_{j \neq i} f_j(x) \end{bmatrix}, \tag{127}$$

where $e_i$ and $e_j$ are one-hot basis, for 2-norm, applying Proposition 2.12 (Lipschitz Constant of Function Concatenation) immediately yields:

$$\mathrm{Lip}\,[m]^2 \leq \mathrm{Lip}\,[f]^2 + \mathrm{Lip}\,[f]^2 \implies \mathrm{Lip}\,[m] \leq \sqrt{2}\,\mathrm{Lip}\,[f]. \tag{128}$$

Therefore, a guaranteed 2-norm perturbation $\delta_x$ does not alter the prediction, if:

$$\|\delta_x\|_2 \leq \frac{m(x)}{\mathrm{Lip}\,[m]} = \frac{m(x)}{\sqrt{2}\,\mathrm{Lip}\,[f]} \tag{129}$$

(also see Tsuzuku et al., 2018, Proposition 1 & 2), which implies that the class margin must be at least:

$$m(x) = f_i(x) - \max_{j \neq i} f_j(x) \geq \sqrt{2}\,c\,\mathrm{Lip}\,[f]. \tag{130}$$

**Lipschitz-Margin Training.** Of the same setting, in training stage, Tsuzuku et al. (2018) adjust each class margin $j \neq i$ by:

$$f_j(x) \leftarrow f_j(x) + \sqrt{2}\,c\,\mathrm{Lip}\,[f], \tag{131}$$

where $c$ is a guaranteed perturbation bound (Tsuzuku et al., 2018, Algorithm 1). For each linear unit $\phi : \mathbb{R}^m \to \mathbb{R}^n$ in the network, and let $u \in \mathbb{R}^m$ be drawn i.i.d. from $\mathcal{N}(0,1)$. Tsuzuku et al. (2018) use a general method for estimating the Lipschitz constant of $\phi$ iteratively by:

$$u \leftarrow \frac{u}{\|u\|_2}, \qquad v \leftarrow \phi(u), \qquad \sigma \leftarrow \|v\|_2, \qquad u \leftarrow \frac{1}{2}\frac{\partial \|v\|_2^2}{\partial u} \tag{132}$$

(Tsuzuku et al., 2018, Theorem 1 & Algorithm 2). At the end of the iteration $T$, the Lipschitz constant of $\phi$ is given by:

$$\mathrm{Lip}\,[\phi] \approx \sigma, \tag{133}$$

almost surely.

## 4.5  Architectural Regularization

Lipschitz continuity can also be constrained through architectural design choices, including activation functions, parameter structures, initialization schemes, and optimization. As a complement to regularization approaches that do not fall into one single category, we survey these approaches in this section.

### 4.5.1  Orthogonalizing Convolution

Parameter orthogonality in neural networks can directly yield 1-Lipschitz continuous (Proposition 2.8) (Lipschitz Constant of Semi-Orthogonal Matrix)). This is because, for a matrix $W \in \mathbb{R}^{m \times n}$, a semi-orthogonal matrix $W$ preserves the 2-norm:

$$W^\top W = I_n \quad \text{or} \quad WW^\top = I_m \quad \implies \quad \mathrm{Lip}\,[W] = 1, \tag{134}$$

if $m \geq n$ and $W^\top W = I_n$, $W$ is an isometric map; if $m < n$ and $WW^\top = I_m$, $W$ is non-expansive, vanishing on the kernel $\ker(W)$ (Proposition 2.8) (Lipschitz Constant of Semi-Orthogonal Matrix)).

**Extending Semi-Orthogonality to Convolution.** Let $W \in \mathbb{R}^{c_{out} \times c_{in} \times n \times n}$ be a convolutional filter where $c_{out}$ is the number of output channels, $c_{in}$ is the number of input channels, and $n \times n$ are the input dimensions. The action on an input $x \in \mathbb{R}^{c_{in} \times n \times n}$ is denoted by:

$$W \circledast x : \mathbb{R}^{c_{in} \times n \times n} \rightarrow \mathbb{R}^{c_{out} \times n \times n}. \tag{135}$$

Using the norm preserving concept for 1-Lipschitz maps on $\ell_2$, for $\forall\ x \in \mathbb{R}^{c_{in} \times n \times n}$, if the 2-norms are preserved:

$$\|W \circledast x\|_2 = \|x\|_2, \tag{136}$$

then the convolutional filter $W$ is referred to as semi-orthogonal (Trockman & Kolter, 2021).

**Filter Orthogonalization via Cayley Transform.** However, convolutional operation is not matrix multiplication, to apply the existing results from matrix theory, Trockman & Kolter (2021) harness the *Convolutional Theorem* (Jain, 1989) for converting the convolutional operation in spatial domain to multiplication in frequency domain by:

$$\mathscr{F}\left[W \circledast x\right][:,i,j] = \mathscr{F}\left[W\right][:,:,i,j]\,\mathscr{F}\left[x\right][:,:,i,j] = \widehat{W}[:,i,j]\widehat{x}[:,i,j], \tag{137}$$

where $\mathscr{F}$ represents Fourier transform, $\widehat{W} = \mathscr{F}\left[W\right]$, $\widehat{x} = \mathscr{F}\left[x\right]$, $[:,:,i,j]$ and $[:,i,j]$ are slicing operations by fixing indices $(i,j)$) (Trockman & Kolter, 2021, Equation 3 & 4, § 4). Note that $\widehat{W}[:,:,i,j]$ is not orthogonal or semi-orthogonal, Trockman & Kolter (2021) use a bijective *Cayley Transform* for deriving an orthogonal matrix $Q[:,:,i,j]$ from $\widehat{W}[:,:,i,j]$ by:

$$Q[:,:,i,j] = (I - A[:,:,i,j])(I + A[:,:,i,j])^{-1}, \tag{138}$$

where:

$$A[:,:,i,j] = \widehat{W}[:,:,i,j] - \widehat{W}[:,:,i,j]^*, \tag{139}$$

and $A[:,:,i,j]$ is referred to as *skew-Hermitian* matrix (Trockman & Kolter, 2021). Conceptually, the convolution on frequency domain is then computed through:

$$\mathscr{F}\left[W \circledast x\right][:,i,j] = Q[:,:,i,j]\,\widehat{x}[:,i,j] \tag{140}$$
$$= (I - A[:,:,i,j])(I + A[:,:,i,j])^{-1}\,\widehat{x}[:,i,j], \tag{141}$$

then $Q[:,:,i,j]\,\widehat{x}[:,i,j]$ is inverted back to spatial domain by:

$$\mathscr{F}^{-1}\left[Q[:,:,i,j]\,\widehat{x}[:,i,j]\right], \tag{142}$$

which leads to the convolution 1-Lipschitz continuous.

### 4.5.2 Orthogonality with Lie Unitary Group

Lezcano-Casado & Martínez-Rubio (2019) introduce a method, stemming from Lie group theory, through the exponential map, for ensuring 1-Lipschitz by construction. Orthogonal constraints networks can improve the robustness and generalization capabilities (Huang et al., 2018; Bansal et al., 2018). For a matrix $A$, if $AA^* = I$, then the matrix $A$ is referred to as a unitary matrix. Unitary matrices allow to solve the exploding or vanish gradients (Arjovsky et al., 2016; 2017). In the sense of Lie algebra, these unitary matrices form a special orthogonal group on field $\mathbb{R}$:

$$SO(n) = \left\{ A \in \mathbb{R}^{n \times n} \mid A^\top A = I \right\} \tag{143}$$

and unitary group on field $\mathbb{C}$:

$$U(n) = \left\{ A \in \mathbb{C}^{n \times n} \mid A^* A = I \right\} \tag{144}$$

(Lezcano-Casado & Martínez-Rubio, 2019, § 3.1). We are interested in converting a parameter matrix $W \in \mathbb{R}^{n \times n}$ to $SO(n)$ or $U(n)$. Lezcano-Casado & Martínez-Rubio (2019) harness the concept of *skew-symmetric matrix* group (Definition 2.7 (Skew-Hermitian & Skew-Symmetric Matrix)) as a bridge:

$$\mathfrak{so}(n) = \{B \in \mathbb{R}^{n \times n} \ : \ B = -B^\top\} \tag{145}$$

and:

$$\mathfrak{u}(n) = \{B \in \mathbb{R}^{n \times n} \ : \ B = -B^*\}. \tag{146}$$

Taking:

$$B = W - W^\top \tag{147}$$

can easily send a parameter matrix $W$ to $\mathfrak{u}(n)$.

**Connecting to Lie Orthogonal Group.** Then there exists an exponential *surjective* map between two groups:

$$\exp : \mathfrak{so}(n) \to SO(n) \tag{148}$$

and:

$$\exp : \mathfrak{u}(n) \to U(n), \tag{149}$$

which is defined as:

$$\exp(B) = I + B + \frac{1}{2}B^2 + \cdots \tag{150}$$

(Lezcano-Casado & Martínez-Rubio, 2019, § 3.2). Then a network $f(x; W)$ is parameterized on the Lie group $SO(n)$ by an algebraic mapping chain:

$$W \in \mathbb{R}^{n \times n} \xrightarrow{B = W - W^\top} B \in \mathfrak{so}(n) \xrightarrow{A = \exp(B)} A \in SO(n), \tag{151}$$

and the optimization problem becomes:

$$\min_A \ f(x; A) \iff \min_B \ f(x; \exp(B)) \tag{152}$$

(Lezcano-Casado & Martínez-Rubio, 2019, § 3.3).

**Optimizing on Lie Orthogonal Group.** The Lie group $SO(n)$ forms a Riemannian manifold, to optimize a function $f$ on the Riemannian manifold $SO(n)$:

$$f : \mathcal{X} \times SO(n) \to \mathbb{R}, \tag{153}$$

where $\mathcal{X}$ is input space, one may use *Riemannian gradient descent* (Absil et al., 2009). Given a point on the manifold $A \in SO(n)$, $\mathcal{T}_A SO(n)$ is the tangent space at $A$ with induced metric, and a gradient $\Omega \in \mathcal{T}_A SO(n)$, then the geodesic update is:

$$A \leftarrow A \exp\left(-\eta \, A^* \, \Omega\right), \tag{154}$$

where $\eta$ is a learning rate $\eta > 0$ (Lezcano-Casado & Martínez-Rubio, 2019). Computing the Riemannian exponential map is expensive; a cheaper first-order approximation, the *Cayley map*, may be used for the update instead:

$$A \leftarrow A \, \phi\left(-\eta \, A^* \, \Omega\right), \tag{155}$$

where $\phi$ is the *Cayley map*:

$$\phi(A) = (I + \frac{1}{2}A)(I - \frac{1}{2}A)^{-1} \tag{156}$$

(Wisdom et al., 2016; Vorontsov et al., 2017). Finally, this induces an update rule on $B \in \mathfrak{so}(n)$:

$$\exp(B) \leftarrow \exp\Big(B - \eta \, \nabla f\,(x; \exp(B))\Big) \tag{157}$$

where $\nabla f\,(x; \exp(B))$ is the usual gradient with Euclidean metric. In a specially designed recurrent neural network (RNN), Lezcano-Casado & Martínez-Rubio (2019) transform the matrix exponential maps skew-symmetric matrices to orthogonal matrices transforming an optimization problem with orthogonal constraints (Lezcano-Casado & Martínez-Rubio, 2019). Particularly, Lezcano-Casado & Martínez-Rubio (2019) use Padé approximants and the scale-squaring trick to compute machine-precision approximations of the matrix exponential and its gradient (Lezcano-Casado & Martínez-Rubio, 2019). As a result, the optimization remains in the group and the Lipschitz constants of the updated matrices remain 1 by design.

### 4.5.3 Lipschitz Continuous Transformers

$L_2$ **Self-Attention Transformer.** Kim et al. have shown that dot-product self-attention is non-globally Lipschitz continuous (Kim et al., 2021, Theorem 3.1.). Referring to equation 25 and equation 26 in Section 2.6, recall that the distance matrix for *query Q* and *key K* in multi-head dot-product attention matrix are computed through their row-wise dot-product:

$$P := \text{Softmax}\Big(\frac{QK^\top}{\sqrt{d_K}}\Big) = \text{Softmax}\Big(\frac{xW_Q(xW_K)^\top}{\sqrt{d_K}}\Big) \propto \exp\Big\{\Big(\frac{xW_Q(xW_K)^\top}{\sqrt{d_K}}\Big)\Big\}, \tag{158}$$

which leads to unbounded Jacobian norm. To solve the issue caused by dot-product attention computation, Kim et al. propose to use $L_2$ distance instead by:

$$P_{ij} \propto \exp\Big\{\Big(\frac{-\|x_i W_Q - x_j W_K\|_2^2}{\sqrt{d_K}}\Big)\Big\}, \tag{159}$$

where $x_i$ and $x_j$ are the $i$-th and $j$-th tokens of $x$, and $P_{ij}$ represents their attention score (Kim et al., 2021, Equation 8).

Kim et al. further show that, for a sequence length $n$, input dimension $d$, and $h$ heads, its 2-norm Lipschitz bound is:

$$\text{Lip}\,[L_2 - \text{Attention}] \leq \frac{\sqrt{n}}{\sqrt{d_K}}\Big[4\phi^{-1}(n-1) + 1\Big]\left(\sqrt{\sum_{h_i} \|W_{Q,h_i}\|_2^2 \, \|W_{V,h_i}\|_2^2}\right)\|W_O\|_2, \tag{160}$$

and the $\infty$-norm Lipschitz bound:

$$\text{Lip}_\infty\,[L_2 - \text{Attention}] \leq \left[4\phi^{-1}(n-1) + \frac{1}{\sqrt{d_K}}\right]\|W_O^\top\|_\infty \, \max_{h_i}\|W_{Q,h_i}\|_\infty\|W_{Q,h_i}^\top\|_\infty \, \max_{h_j}\|W_{V,h_j}^\top\|_\infty, \tag{161}$$

where:

$$\phi(x) = x\exp(x+1)$$

is an invertible univariate function on $x > 0$, $W_{Q,h_i}$ is the *query* parameter matrix for head $h_i$, $W_{V,h_i}$ is the *value* parameter matrix for head $h_i$, and $W_O \in \mathbb{R}^{d \times d}$ is the projection parameter matrix; the per-head key dimension is $d_K = d/h$ (Kim et al., 2021, Theorem 3.2). They further conclude that their results can lead that the 2-norm Lipschitz bound is at the scale:

$$\text{Lip}[\text{Attention}] \sim O(\sqrt{n}\log n), \tag{162}$$

and the $\infty$-norm Lipschitz bound is at the scale:

$$\mathrm{Lip}_\infty[\text{Attention}] \sim O(\log n) \tag{163}$$

(Kim et al., 2021, Theorem 3.2). The bounds in (Kim et al., 2021, Theorem 3.2) are complemented by concurrent work (Vuckovic et al., 2020) with a measure-theoretical framework. Vuckovic et al. show that the 1-norm Lipschitz bound is at the scale:

$$\mathrm{Lip}_1[\text{Attention}] \sim O(\sqrt{\log n}) \tag{164}$$

(Vuckovic et al., 2020, Theorem 29 & Corollary 30).

*Remark* 4.1. The Lipschitz constant bound of an $L_2$ self-attention layer remains dependent on the sequence length $n$ but is independent of the input norm.

**LipsFormer.** Qi et al. (2023) demonstrate that enforcing Lipschitz continuity in Transformer architecture (Vaswani et al., 2017) is more crucial for ensuring training stability in contrast to the tricks such as *learning rate warmup*, *layer normalization*, *attention formulation*, and *weight initialization* (Qi et al., 2023). Their proposed method, referred to as **LipsFormer**, replaces the Transformer parts with their Lipschitz continuous counterparts: *CenterNorm* for *LayerNorm*, *spectral initialization* for *Xavier initialization*, *scaled cosine similarity attention* for *dot-product attention*, and *weighted residual shortcut* for *residual connection* (Qi et al., 2023). These modifications result in a Transformer architecture with a provably bounded Lipschitz constant. Their key architectural design elements are summarized as follows:

1. **CenterNorm instead of LayerNorm.** LayerNorm (Ba et al., 2016) is widely used in Transformer, defined as:

$$\mathrm{LN}(x) = \gamma \odot z + \beta, \tag{165}$$

   and:

$$z = \frac{y}{\mathrm{std}(y)}, \qquad y = \left(I_d - \frac{1}{d}\mathbf{1}_d\mathbf{1}_d^\top\right)x, \tag{166}$$

   where $x \in \mathbb{R}^d$ is input, $I_d$ is an identity matrix in $\mathbb{R}^{d\times d}$, $\mathbf{1}_d$ is an all-ones vector in $\mathbb{R}^d$, $\mathrm{std}(y)$ is the standard deviation of $y$, $\odot$ is element-wise product, $\gamma$ and $\beta$ are learnable parameters initialized to 1 and 0 respectively (Qi et al., 2023). The Jacobian of $z$ with respect to $x$ is given as:

$$\frac{\partial z}{\partial x} = \frac{1}{\mathrm{std}(y)}\left(I_d - \frac{1}{d}\mathbf{1}_d\mathbf{1}_d^\top\right)\left(I_d - \frac{yy^\top}{\|y\|_2^2}\right), \tag{167}$$

   which shows that **LayerNorm** is not Lipschitz continuous when $\mathrm{std}(y) \to 0$. To address this problem, Qi et al. (2023) introduce **CenterNorm**, defined as:

$$\mathrm{CN}(x) = \gamma \odot \frac{d}{d-1}\left(I_d - \frac{1}{d}\mathbf{1}_d\mathbf{1}_d^\top\right)x + \beta, \tag{168}$$

   which admits a Lipschitz constant:

$$\mathrm{Lip}\,[\mathrm{CN}] = \frac{d}{d-1} \approx 1 \tag{169}$$

   for sufficient large $d$, $\gamma = 1$, and $\beta = 0$ (Qi et al., 2023).

2. **Scaled Cosine Similarity Attention (SCSA).** Kim et al. (2021) show that standard dot-product self-attention is not Lipschitz continuous since the Lipschitz constant depends on sequence length which is not bounded (Kim et al., 2021), see also Section 2.6 (Lipschitz Continuity of Dot-Product Self-Attention). To address this problem, Qi et al. (2023) replace standard dot-product attention part with **SCSA**, defined as:

$$\mathrm{SCSA}(x;\nu,\tau) = \nu PV, \qquad P = \mathrm{softmax}(\tau QK^\top), \tag{170}$$

   where $\nu$ and $\tau$ are learnable scalars, and $K, Q, V$ are row-wise normalized in $\ell_2$ (Qi et al., 2023, § 4.1.2).

*Remark* 4.2. It is worth noting that their method reduces the Lipschitz constant of the attention module; however, the module remains not globally Lipschitz continuous since the Lipschitz constant still depends on sequence length $n$:

$$\text{Lip}_2\left[SCSA\right] \leq O(n^2), \qquad \text{and} \qquad \text{Lip}_\infty\left[SCSA\right] \leq O(n^2) \tag{171}$$

(Qi et al., 2023, Theorem 1).

3. **Weighted Residual Shortcut.** For a residual module:

$$h(x) = x + g(x), \tag{172}$$

where $g : \mathbb{R}^d \to \mathbb{R}^d$, $h : \mathbb{R}^d \to \mathbb{R}^d$ and $x \in \mathbb{R}^d$, the Lipschitz constant of $h$ is bounded by:

$$\text{Lip}\left[h\right] \leq 1 + \text{Lip}\left[g\right], \tag{173}$$

see Section 2.7.3 (Lipschitz Constant of Residual Network). To further reduce the Lipschitz constant introduced by residual module, Qi et al. (2023) replace the standard residual module with:

$$\text{WRS}(x) = x + \alpha \odot g(x), \tag{174}$$

where $\alpha$ is a learnable parameter, initialized to a small value $[0.1, 0.2]$ (Qi et al., 2023). Assuming the residual branch $g$ is 1-Lipschitz, the Lipschitz constant of WRS in residual module is bounded by:

$$\text{Lip}\left[\text{WRS}\right] \leq 1 + \max(\alpha). \tag{175}$$

4. **Spectral-based Weight Initialization.** Qi et al. (2023) further normalize the parameters at initialization by spectral-norm. Let $W$ be a parameter matrix, the $W$ is normalized by:

$$W \leftarrow \frac{W}{\|W\|_2} \tag{176}$$

(Qi et al., 2023, § 4.1.4).

### 4.5.4 Jacobian Norm Minimization

Constraining the Lipschitz constant of Transformer architecture for stable optimization and robustness remains an open research problem. Yudin et al. (2025) first derive an explicit tight local Lipschitz constant for self-attention module in Transformer architecture by analyzing the Jacobian of softmax (Yudin et al., 2025). Building on top of the Jacobian analysis, they present **JaSMin** (**Ja**cobian **S**oftmax norm **Min**imization) for enhancing Transformer's robustness by constraining the local Lipschitz constant (Yudin et al., 2025).

**Bounding Lipschitz through Softmax Jacobian.** Consider the softmax function:

$$\text{softmax} : \mathbb{R}^d \to \mathbb{R}^d, \tag{177}$$

then the Jacobian of the softmax function is:

$$\mathcal{M}(P_x) := \nabla \text{softmax}\,(x) = \text{diag}(P_x) - P_x P_x^\top, \tag{178}$$

where:

$$P_x = \left(\frac{e^{x_i}}{\sum_j e^{x_j}}\right). \tag{179}$$

Yudin et al. (2025) incorporate this Jacobian expression of the softmax function into self-attention, leading to a Lipschitz constant bound for a single head:

$$\text{Lip}\left[\text{Attention}\right] \leq \|W_V\|_2\left(\|P\|_2 + 2\|x\|_2^2\,\|A\|_2\,\max_i \|\mathcal{M}(P_{i,:})\|_2\right) \tag{180}$$

with:

$$P = \text{softmax}\left(xAx^\top\right) \in \mathbb{R}^{n \times n} \quad \text{and} \quad A = \frac{W_Q W_K^\top}{\sqrt{d_K}} \in \mathbb{R}^{d \times d} \tag{181}$$

(Yudin et al., 2025, Theorem 3). It is worth noting that the local Lipschitz constant contains a term:

$$\text{Lip}\Big[\text{Attention}\Big] \leq O\Big(\max_i \|\mathcal{M}(P_{i,:})\|_2\Big), \tag{182}$$

which shows that softmax Jacobian norm determines an upper bound of the local Lipschitz constant.

**Jacobian Softmax Norm Minimization.** To approximate the Jacobian norm more efficiently, Yudin et al. (2025) introduce a surrogate function $g$ that captures the partial ordering of the singular values of softmax mappings. For $x \in \mathbb{R}^n_{>0}$, let $x_{(k)}$ be the $k$-th largest component of $x$. Define:

$$g_k(x) := x_{(k)}(1 - x_{(k)} + x_{(k+1)}), \tag{183}$$

for $k = 1, 2, \cdots, n-1$ (Yudin et al., 2025, Definition 1). Let $A = \text{diag}(x) - xx^\top$, then the following inequality holds true:

$$x_{(1)} \geq g_1(x) \geq \sigma_1(A) \geq x_{(2)} \geq g_2(x) \geq \sigma_2(A) \geq x_{(n)} \geq g_n(x) \geq \sigma_n(A) \geq 0 \tag{184}$$

(Yudin et al., 2025, Theorem 4).

Building on the derived bound, Yudin et al. (2025) propose an efficient regularization loss expressed in terms of the function $g$ with two forms:

$$\mathcal{L}_{\text{JaSMin}(k=0)} = \sum_{\ell=1}^{L} \sum_{h_i=1}^{h} \max_j \log\Big[g_1\Big(P_{j,:}^{\ell,i}\Big)\Big], \tag{185}$$

and:

$$\mathcal{L}_{\text{JaSMin}(k)} = \sum_{\ell=1}^{L} \sum_{h_i=1}^{h} \max_j \log\left[\frac{g_1\Big(P_{j,:}^{\ell,i}\Big)}{g_k\Big(P_{j,:}^{\ell,i}\Big)}\right], \tag{186}$$

where $P_{j,:}^{\ell,i}$ represents the softmax map for the $\ell$-th attention module ($1 \leq \ell \leq L$), $i$-th head ($1 \leq i \leq h$), and the $j$-th row of the map (Yudin et al., 2025, § 4). They also show that for $g_1/g_k$ is bounded below $\gamma$:

$$\frac{g_1\Big(P_{j,:}^{\ell,i}\Big)}{g_k\Big(P_{j,:}^{\ell,i}\Big)} \leq \gamma, \tag{187}$$

for $1 \leq \gamma \leq \frac{k}{4}$. Then the softmax Jacobian 2-norm is bounded by:

$$\|\mathcal{M}(P_{j,:}^{\ell,i})\|_2 \leq O\Big(\frac{\gamma}{k}\Big) \tag{188}$$

(Yudin et al., 2025, Proposition 1).

### 4.6 Remarks

The regularization approaches surveyed above also involve a trade-off between enforceability, computational cost, expressive power, and guarantee strength. Weight-based methods such as weight clipping (Arjovsky et al., 2017) (see Section 4.1) and spectral normalization (Miyato et al., 2018; Yoshida & Miyato, 2017) (see Section 4.1) are simple to implement and scale well to large models, but they typically control only upper bounds induced by layer norms rather than the exact network Lipschitz constant (see Proposition 2.17 and

Proposition 2.19). Gradient-based methods (Gulrajani et al., 2017) (see Section 4.2) act more directly on the input–output sensitivity by penalizing gradient norms (see Lemma 2.5), yet they are often local in nature and can be computationally expensive due to repeated differentiation. Activation- and architecture-based approaches, such as GroupSort (Anil et al., 2019), invertible residual constructions (Behrmann et al., 2019; Perugachi-Diaz et al., 2021), and Lipschitz-continuous transformer variants (Kim et al., 2021; Qi et al., 2023; Yudin et al., 2025) (see Sections 4.3 and 4.5), provide stronger structural control by design, but may restrict architectural flexibility or reduce expressive efficiency. Class-margin regularization (Tsuzuku et al., 2018) (see Section 4.4) connects Lipschitz control to certifiable robustness more directly through margin-based objectives, although its effectiveness still depends on the quality of the underlying Lipschitz bound (see Equation equation 129).

## 5 Certifiable Robustness

Certifiable robustness, a cornerstone of trustworthy deep learning, guarantees that a neural network's predictions remain consistent under adversarial perturbations within a specified norm ball. Unlike empirical defenses like adversarial training, which lack formal guarantees, Lipschitz-based certification methods leverage the Lipschitz constant to bound sensitivity to input changes, ensuring provable robustness. While some literature, such as (Clevert et al., 2016), have been introduced previously from the perspectives — such as **theoretical foundations**, **estimation methods** and **regularization approaches**, we may revisit some of them in this section from the perspective of **certifiable robustness**.

### 5.1 Formal Robustness Guarantees

To complement the results in Section 4.4, we now discuss formal robustness guarantees for the general case. In the setting of a $k$-class classifier $f$ with Lipschitz constant $K$, a central question in adversarial defense is to determine the largest perturbation norm that leaves the prediction unchanged.

**Theorem 5.1** ($p$-Norm Lipschitz Margin Robustness Radius). *Let $f : \mathbb{R}^d \to \mathbb{R}^k$ be a classifier. Let $f_i$ be the $i$-th output. Let $c = \arg\max_i f_i(x)$ be the predicted class for $x$. Define the (one-versus-rest) margin*

$$m(x) := f_c(x) - \max_{j \neq c} f_j(x). \tag{189}$$

*Let $f$ be Lipschitz continuous from the input $\ell_p$ norm to the output $p$-norm,* i.e.,

$$\|f(x) - f(y)\|_p \leq \mathrm{Lip}_p[f] \|x - y\|_p \quad \text{for all } x, y. \tag{190}$$

*Then for any perturbation $\delta_x$ with*

$$\|\delta_x\|_p < \frac{m(x)}{2^{1-\frac{1}{p}} \mathrm{Lip}_p[f]}, \tag{191}$$

*the prediction is unchanged:*

$$\arg\max_i f_i(x + \delta_x) = c. \tag{192}$$

*Remark* 5.2. This result is particularly useful for analyzing the adversarial defense problem. Similar results have been discussed in the literature (Szegedy et al., 2014; Hein & Andriushchenko, 2017; Cisse et al., 2017; Tsuzuku et al., 2018).

*Proof.* By writing equation 126 into:

$$m(x) = (e_c - e_j)^\top \begin{bmatrix} f_c(x) \\ \max_{j \neq c} f_j(x) \end{bmatrix},$$

where $e_c$ and $e_j$ are one-hot basis, for $p$-norm, applying Hölder's inequality with the conjugate exponent $q$ ($\frac{1}{p} + \frac{1}{q} = 1$) immediately yields:

$$\mathrm{Lip}_p\,[m] \leq \|e_c - e_j\|_q \;\mathrm{Lip}_p\,[f] = 2^{\frac{1}{q}}\,\mathrm{Lip}_p\,[f] = 2^{1-\frac{1}{p}}\,\mathrm{Lip}_p\,[f],$$

so that a guaranteed $p$-norm perturbation $\delta_x$ does not alter the prediction, if:

$$\|\delta_x\|_p \leq \frac{m(x)}{\mathrm{Lip}_p\,[m]} = \frac{m(x)}{2^{1-\frac{1}{p}}\,\mathrm{Lip}_p\,[f]},$$

which recovers equation 129 in Section 4.4 with $p = 2$.

$\square$

## 5.2 Certifiable Robustness via Global Lipschitz Bound

For a network $f$ with 1-Lipschitz activations and $W^{(\ell)}$ is the $\ell$-th layer parameter matrix, the spectral-norm product bounds the Lipschitz constant of $f$ globally:

$$\mathrm{Lip}[f] \leq \prod_{\ell=1}^{L} \|W^{(\ell)}\|_2,$$

providing a *certified global bound*. This is computationally cheap but typically loose. **SeqLip** (Virmaux & Scaman, 2018) refines this spectral product by incorporating singular-vector alignment between layers, also producing certified global bounds. **LipSDP** (Fazlyab et al., 2019) interprets activation functions as gradients of convex potential functions, satisfying certain properties described by quadratic constraints. Therefore, Lipschitz constant estimation problem is treated as a semi-definite program (SDP) problem.

## 5.3 Certifiable Robustness via Local Lipschitz Bound

Local Lipschitz bounds are computed over a neighborhood. **Fast-Lip** (Weng et al., 2018a) analyze the activation patterns of ReLU networks and derive a gradient norm based local Lipschitz bound. **CLEVER** (Weng et al., 2018b) estimates local constants via extreme value theory on sampled gradient norms; this improves tightness but does not yield a formal certificate. MILP-based methods (Jordan & Dimakis, 2020) encode the exact piecewise-linear constraints of ReLU networks within the ball, yielding exact local Lipschitz constants (and thus exact certified radii), but are limited to small networks.

## 5.4 Certifiable Robustness in Language Models

**Proper Distance Metrics for Text.** Language models typically operate on text sequences as inputs, for which the conventional notion of a Lipschitz constant over $\mathbb{R}$ with $\ell_p$ norm does not directly apply. The *Levenshtein distance* (Levenshtein, 1966) measures the number of character replacements, insertions or deletions needed in order to transform a sequence $x$ to sequence $y$. It is a proper metric that satisfies all axioms of a metric space:

1. **Non-Negativity.** $d_{\mathrm{lev}}(x, y) \geq 0$;

2. **Identity.** $d_{\mathrm{lev}}(x, y) = 0 \iff x = y$;

3. **Symmetry.** $d_{\mathrm{lev}}(x, y) = d_{\mathrm{lev}}(y, x)$;

4. **Triangle Inequality.** $d_{\mathrm{lev}}(x, z) \leq d_{\mathrm{lev}}(x, y) + d_{\mathrm{lev}}(y, z)$;

where $d_{\mathrm{lev}}(x, y)$ denotes the Levenshtein distance between two sequences $x$ and $y$. Thereby it provides a principled foundation for discussing Lipschitz continuity in the context of language models. For tokenized text — where each vocabulary item is represented as a one-hot vector — a variant of the *Levenshtein distance*,

known as the *ERP distance*, has been proposed for comparing sequences of one-hot vectors (Chen & Ng, 2004).

Rocamora et al. (2024) introduce **LipsLev**, a method for training convolutional text classifiers that are *1-Lipschitz* in the sense of Levenshtein distance for Lipschitz constant based defense (Rocamora et al., 2024). That is, for a classifier $f : S \to \mathbb{R}^c$, two one-hot vector-valued sequences $x \in \mathbb{R}^{n \times d} \subset S$ and $y \in \mathbb{R}^{m \times d} \subset S$ where $n, m$ are sequence lengths and $d$ is one-hot vector dimension, the classification margin must hold:

$$|f_i(x) - f_j(y)| \leq K_{\text{lev}} \, d_{\text{lev}}(x, y) \tag{193}$$

for a constant $K_{\text{lev}} \leq 1$, so that the prediction does not change under perturbation radius $d_{\text{lev}}(x, y) < R$ (Rocamora et al., 2024, Theorem 4.3 & Corollary 4.4). They then train a 1-Lipschitz classifier over text sequences — in the sense of the Levenshtein distance — by normalizing each layer's outputs through division by its corresponding Lipschitz constant (Rocamora et al., 2024, § 4.3).

## 6  Conclusions

Lipschitz continuity serves as a cornerstone for enhancing the robustness, generalization, and optimization stability of deep neural networks, providing a rigorous mathematical framework to quantify and control sensitivity to input perturbations. This comprehensive survey synthesizes key insights across theoretical foundations, estimation techniques, regularization approaches, and certifiable robustness, highlighting their interconnections and practical implications. By unifying disparate research threads, we address a critical gap in the literature, offering a cohesive resource for researchers and practitioners. Challenges remain in balancing bound tightness with computational scalability, preserving model expressivity under constraints, and extending certifications to diverse norms and large-scale architectures like transformers, paving the way for future advancements in trustworthy deep learning systems.

## Acknowledgments

This publication has emanated from research conducted with the financial support of **Taighde Éireann** – Research Ireland under grant number 18/CRT/6223. The authors gratefully acknowledge the partial in-kind support provided by the J.E. Cairnes School of Business & Economics at the University of Galway, Ireland. We especially thank Ping Song of Research Ireland – Centre for Research Training in AI & the University of Galway (Ireland) for the help. For the purpose of Open Access, the author has applied a CC BY public copyright license to any Author Accepted Manuscript version arising from this submission. We also thank reviewers for their constructive comments which can significantly improve our research quality.

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

# A Proofs: Lipschitz Constants of Activation Functions

## A.1 Proof: Lipschitz Constant of Sigmoid

*Proof.* Consider the sigmoid function:

$$\text{Sigmoid}(x) = f(x) = (1 + e^{-x})^{-1}. \tag{194}$$

Let $u = 1 + e^{-x}$, so $f(x) = u^{-1}$. The derivative is:

$$f'(x) = -\frac{1}{u^2} \cdot \frac{du}{dx}. \tag{195}$$

Compute $\frac{du}{dx}$:

$$u = 1 + e^{-x}, \quad \frac{du}{dx} = -e^{-x}. \tag{196}$$

Thus:

$$f'(x) = -\frac{1}{(1 + e^{-x})^2} \cdot (-e^{-x}) = \frac{e^{-x}}{(1 + e^{-x})^2}. \tag{197}$$

Express the derivative in terms of $f(x)$:

$$1 - f(x) = \frac{e^{-x}}{1 + e^{-x}}, \tag{198}$$

and:

$$f'(x) = \frac{1}{1 + e^{-x}} \cdot \frac{e^{-x}}{1 + e^{-x}} = f(x)(1 - f(x)). \tag{199}$$

Thus:

$$|f'(x)| = f(x)(1 - f(x)). \tag{200}$$

**Maximize the Derivative.** Since $0 < f(x) < 1$, we maximize $g(z) = z(1 - z)$ for $z = f(x) \in (0, 1)$:

$$g'(z) = 1 - 2z = 0 \implies z = \frac{1}{2}, \tag{201}$$

thus:

$$g\left(\frac{1}{2}\right) = \frac{1}{2} \cdot \frac{1}{2} = \frac{1}{4}. \tag{202}$$

Find when $f(x) = \frac{1}{2}$:

$$\frac{1}{1 + e^{-x}} = \frac{1}{2} \implies e^{-x} = 1 \implies x = 0, \tag{203}$$

at $x = 0$:

$$f'(0) = \frac{1}{2} \cdot \frac{1}{2} = \frac{1}{4}. \tag{204}$$

As $x \to \infty$, $e^{-x} \to 0$, so $f'(x) \to 0$. As $x \to -\infty$, $e^{-x} \to \infty$, so:

$$f'(x) \approx \frac{e^{-x}}{e^{-2x}} = e^x \to 0. \tag{205}$$

Hence:

$$\mathrm{Lip}\left[\mathrm{Sigmoid}(x)\right] = \sup_x |f'(x)| = \frac{1}{4}. \tag{206}$$

$\square$

### A.2 Proof: Lipschitz Constant of Tanh

*Proof.* The tanh is defined as:

$$f(x) = \tanh(x) = \frac{e^x - e^{-x}}{e^x + e^{-x}}, \tag{207}$$

for all $x, y \in \mathbb{R}$.

Let $g(x) = \sinh(x)$, $h(x) = \cosh(x)$, so $f(x) = \frac{g(x)}{h(x)}$. The derivative is:

$$f'(x) = \frac{g'(x)h(x) - g(x)h'(x)}{h(x)^2}. \tag{208}$$

Since $g'(x) = \cosh(x)$, $h'(x) = \sinh(x)$, and $h(x)^2 = \cosh^2(x)$:

$$f'(x) = \frac{\cosh(x) \cdot \cosh(x) - \sinh(x) \cdot \sinh(x)}{\cosh^2(x)} = \frac{\cosh^2(x) - \sinh^2(x)}{\cosh^2(x)}. \tag{209}$$

Using the identity $\cosh^2(x) - \sinh^2(x) = 1$:

$$f'(x) = \frac{1}{\cosh^2(x)} = \mathrm{sech}^2(x). \tag{210}$$

Since $\cosh(x) \geq 1$, we have:

$$|f'(x)| = \mathrm{sech}^2(x). \tag{211}$$

Express the derivative in terms of $f(x)$:

$$f'(x) = \mathrm{sech}^2(x) = 1 - \tanh^2(x) = 1 - f(x)^2. \tag{212}$$

Since $-1 \leq f(x) \leq 1$, we have $|f'(x)| = 1 - f(x)^2 \leq 1$.

**Maximize the Derivative.** Maximize $|f'(x)| = \mathrm{sech}^2(x)$:

$$\cosh(x) = \frac{e^x + e^{-x}}{2}. \tag{213}$$

At $x = 0$:

$$\cosh(0) = \frac{e^0 + e^0}{2} = 1, \quad \mathrm{sech}^2(0) = \frac{1}{\cosh^2(0)} = 1. \tag{214}$$

As $|x| \to \infty$, $\cosh(x) \approx \frac{e^{|x|}}{2}$, so:

$$\text{sech}^2(x) \approx \frac{4}{e^{2|x|}} \to 0. \tag{215}$$

The supremum of $|f'(x)|$ is 1 at $x = 0$. Alternatively, since $f(x)^2 \le 1$, the supremum of $1 - f(x)^2$ occurs when $f(x) = 0$:

$$\tanh(0) = 0, \quad f'(0) = 1 - 0^2 = 1. \tag{216}$$

Hence:

$$\text{Lip}\,[\tanh(x)] = \sup_x |f'(x)| = 1. \tag{217}$$

$\square$

### A.3 Proof: Lipschitz Constant of Softplus

*Proof.* Consider the softplus function:

$$f(x) = \ln(1 + e^x). \tag{218}$$

The derivative is:

$$f'(x) = \frac{d}{dx} \ln(1 + e^x) = \frac{e^x}{1 + e^x}. \tag{219}$$

Since $e^x > 0$ and $1 + e^x > 1$, we have $f'(x) > 0$, so:

$$|f'(x)| = \frac{e^x}{1 + e^x}. \tag{220}$$

**Maximize the Derivative.** To find the Lipschitz constant, we need to compute:

$$K = \sup_{x \in \mathbb{R}} \frac{e^x}{1 + e^x}. \tag{221}$$

Notice that $\frac{e^x}{1+e^x}$ is the sigmoid function, which ranges between 0 and 1. Analyze its behavior:

1. As $x \to \infty$, $e^x$ grows large, so:

$$\frac{e^x}{1 + e^x} \approx \frac{e^x}{e^x} = 1. \tag{222}$$

2. As $x \to -\infty$, $e^x \to 0$, so:

$$\frac{e^x}{1 + e^x} \to \frac{0}{1} = 0. \tag{223}$$

To confirm the supremum, consider the function $g(x) = \frac{e^x}{1+e^x}$. Its derivative is:

$$g'(x) = \frac{e^x(1 + e^x) - e^x \cdot e^x}{(1 + e^x)^2} = \frac{e^x}{(1 + e^x)^2}. \tag{224}$$

Since $g'(x) > 0$ for all $x$, $g(x)$ is strictly increasing, approaching 0 as $x \to -\infty$ and 1 as $x \to \infty$. Thus:

$$\sup_{x \in \mathbb{R}} \frac{e^x}{1 + e^x} = 1 \tag{225}$$

Hence:

$$\mathrm{Lip}\left[\mathrm{Softplus}(x)\right] = \sup_x |f'(x)| = 1. \tag{226}$$

$\square$

### A.4 Proof: Lipschitz Constant of Swish

*Proof.* Let

$$f(x) = x\,\sigma(x), \quad \sigma(x) = \frac{1}{1 + e^{-x}}. \tag{227}$$

The derivative is

$$g(x) := f'(x) = \sigma(x) + x\,\sigma(x)\big(1 - \sigma(x)\big). \tag{228}$$

Since $\sigma(-x) = 1 - \sigma(x)$, we have

$$g(-x) = 1 - g(x). \tag{229}$$

Using $\sigma(x) = \frac{1}{2}\big(1 + \tanh\big(\frac{x}{2}\big)\big)$ and $\sigma(x)(1 - \sigma(x)) = \frac{1}{4}\,\mathrm{sech}^2\big(\frac{x}{2}\big)$,

$$g(x) = \frac{1}{2}\Big(1 + \tanh\frac{x}{2}\Big) + \frac{x}{4}\,\mathrm{sech}^2\frac{x}{2}. \tag{230}$$

Differentiating,

$$g'(x) = \frac{1}{4}\,\mathrm{sech}^2\frac{x}{2}\Big(2 - x\,\tanh\frac{x}{2}\Big), \tag{231}$$

so the maximizer $x^\star > 0$ satisfies

$$x^\star\,\tanh\frac{x^\star}{2} = 2. \tag{232}$$

Using $\tanh\big(\frac{x^\star}{2}\big) = \frac{2}{x^\star}$ and

$$\mathrm{sech}^2\frac{x^\star}{2} = \frac{(x^\star)^2 - 4}{(x^\star)^2}. \tag{233}$$

Thus

$$K = g(x^\star) = \frac{1}{2} + \frac{x^\star}{4}. \tag{234}$$

Since $g(-x) = 1 - g(x)$, $x^\star$ gives the global maximum of $|g|$, solving $x^\star\,\tanh\frac{x^\star}{2} = 2$ gives:

$$x^\star \approx 2.3993572805\cdots. \tag{235}$$

Hence:

$$\mathrm{Lip}\left[\mathrm{Swish}(x)\right] = \sup_x |f'(x)| \approx 1.09983932\cdots \tag{236}$$

$\square$

### A.5 Proof: Lipschitz constant of GELU

*Proof.* Let:

$$f(x) = x\,\Phi(x), \tag{237}$$

where:

$$\Phi(x) = \tfrac{1}{2}\big(1 + \mathrm{erf}(x/\sqrt{2})\big) \tag{238}$$

is the standard normal CDF and:

$$\phi(x) = \tfrac{1}{\sqrt{2\pi}}e^{-x^2/2} \tag{239}$$

is its PDF.

Differentiate:

$$g(x) := f'(x) = \Phi(x) + x\,\phi(x). \tag{240}$$

Note the symmetry $\Phi(-x) = 1 - \Phi(x)$ and $\phi(-x) = \phi(x)$, hence

$$g(-x) = 1 - g(x). \tag{241}$$

Compute the critical points:

$$g'(x) = \phi(x) + \phi(x) + x\,\phi'(x) = 2\phi(x) - x^2\phi(x) = \phi(x)\,(2 - x^2). \tag{242}$$

Since $\phi(x) > 0$, we have $g'(x) > 0$ for $|x| < \sqrt{2}$ and $g'(x) < 0$ for $|x| > \sqrt{2}$. Hence $g$ attains its unique global maximum at $x^\star = \sqrt{2}$ and minimum at $-\sqrt{2}$.

Since $g(-x) = 1 - g(x)$, the minimum equals $1 - g(\sqrt{2})$, so indeed $\sup_x |g(x)| = g(\sqrt{2})$.

Evaluate at:

$$x = \sqrt{2} \approx 1.414214\cdots, \tag{243}$$

hence:

$$\mathrm{Lip}\,[\mathrm{GELU}(x)] = \sup_x |f'(x)| \tag{244}$$

$$= g(\sqrt{2}) \tag{245}$$

$$= \Phi(\sqrt{2}) + \sqrt{2}\,\phi(\sqrt{2}) \tag{246}$$

$$= \tfrac{1}{2}\big(1 + \mathrm{erf}(1)\big) + \frac{e^{-1}}{\sqrt{\pi}} \tag{247}$$

$$\approx 1.128904145. \tag{248}$$

$$\square$$

### A.6 Proof: Lipschitz constant of Softmax

*Proof.* Let

$$\mathrm{softmax}(z)_i = \frac{e^{z_i}}{\sum_{j=1}^n e^{z_j}}, \tag{249}$$

and:

$$p := \mathrm{softmax}(z) \in \Delta^{n-1}, \tag{250}$$

where $\Delta^{n-1}$ is the probability simplex in $\mathbb{R}^n$ (*i.e.*, the set of all probability vectors of length $n$):

$$\Delta^{n-1} := \big\{\, p \in \mathbb{R}^n \mid p_i \geq 0, \; \sum_{i=1}^{n} p_i = 1 \,\big\}. \tag{251}$$

The Jacobian is

$$J(z) = \nabla\,\mathrm{softmax}(z) = \mathrm{diag}(p) - p\,p^\top, \tag{252}$$

which is symmetric positive semidefinite (PSD) and satisfies:

$$J(z)\,\mathbf{1} = 0. \tag{253}$$

Hence the Lipschitz constant $K$ is:

$$K = \sup_{z \in \mathbb{R}^n} \|J(z)\|_2 = \sup_{p \in \Delta^{n-1}} \lambda_{\max}\big(\mathrm{diag}(p) - p\,p^\top\big). \tag{254}$$

where $\lambda_{\max}$ is the largest eigenvalue of $\mathrm{diag}(p) - p\,p^\top$.

For any unit vector $v \in \mathbb{R}^n$,

$$v^\top J(z)\,v = v^\top \mathrm{diag}(p)\,v - (p^\top v)^2 = \sum_{i=1}^{n} p_i v_i^2 - \Big(\sum_{i=1}^{n} p_i v_i\Big)^2 = \mathrm{Var}_{i \sim p}[v_i]. \tag{255}$$

Therefore

$$\|J(z)\|_2 = \sup_{\|v\|_2 = 1} \mathrm{Var}_{i \sim p}[v_i]. \tag{256}$$

By Popoviciu's inequality on variances, for any random variable $X$ supported in $[a, b]$,

$$\mathrm{Var}[X] \leq \frac{(b-a)^2}{4}, \tag{257}$$

with equality attained by a two–point distribution at the endpoints.

Applying this to $X = v_i$ under $p$, we get

$$\mathrm{Var}_{i \sim p}[v_i] \leq \frac{\big(\max_i v_i - \min_i v_i\big)^2}{4}. \tag{258}$$

Maximizing the RHS over $\|v\|_2 = 1$ yields $\max_i v_i - \min_i v_i \leq \sqrt{2}$, with equality for

$$v = \frac{1}{\sqrt{2}}(e_a - e_b) \quad \text{for some distinct } a, b, \tag{259}$$

so that

$$\sup_{\|v\|_2 = 1} \mathrm{Var}_{i \sim p}[v_i] \leq \frac{(\sqrt{2})^2}{4} = \frac{1}{2}. \tag{260}$$

This upper bound is (arbitrarily) attainable by choosing $p$ supported on the two indices $a, b$ with $p_a = p_b = \frac{1}{2}$ and $p_i \to 0$ for $i \notin \{a, b\}$ (which is approached by softmax logits $z_a = z_b \gg z_{i \notin \{a,b\}}$). In that case,

$$J = \begin{bmatrix} \frac{1}{4} & -\frac{1}{4} \\ -\frac{1}{4} & \frac{1}{4} \end{bmatrix} \oplus 0_{n-2}, \tag{261}$$

whose largest eigenvalue is $\frac{1}{2}$.

Hence:

$$\mathrm{Lip}\left[\mathrm{Softmax}(x)\right] = \sup_z \|J(z)\|_2 = \frac{1}{2}, \tag{262}$$

independent of $n \geq 2$.

$\square$

