# OpenReview forum: "Lipschitz Continuity in Deep Learning: A Systematic Review of Theoretical Foundations, Estimation Methods, Regularization Approaches, and Certifiable Robustness"
_TMLR — Accepted by TMLR_

### Review · Reviewer_X65t · 2025-11-05

**Summary Of Contributions:**

This survey paper consolidates findings regarding Lipschitz continuity in deep learning. Specifically, the authors cover four main areas of study: theoretical foundations, estimation methods, regularization approaches, and certifiable robustness. Each section details several relevant approaches / mathematical results.

Strengths:

S1. The paper is generally well-written, with logically grouped sections, consistent mathematical notation, and clear prose.

S2. The authors provide a few novel theoretical results: Theorem 2.9, which provides a Lipschitz bound for a feedforward DAG, and Appendix A.6 which proves that the Lipschitz constant of softmax is 1/2 instead of 1 (this is also shown in concurrent work [1]). I haven't seen Theorem 2.9 before in the literature.

S3. The paper is fairly thorough in its survey of Lipschitzness in deep learning.

Weaknesses:

W1. Some sections read like a lists of theorem statements and results (e.g. Sec 2.4). While the content itself is good, I'd expect more context before / after theorem statements.

W2. Lipschitz continuity analyses have not been shown to scale to deep / heavily overparameterized models, making me skeptical of their applicability for modern ML approaches.

W3. Theorem 2.9 provides a pretty loose / naive bound. Consider for example, a graph a -> b -> c -> d -> e -> f -> g with skip connections a -> c and e -> g (such skip connections are very common). The tightest Lipschitz bound would arise from bounding the a through d part of the network and then the d through g part of the network. In this result, the Lipschitz bound scales combinatorially in the total number of paths.

W4. The only exception to S3 above is I wish there were more discussion of designed Lipschitz architectures such as [2].

W5. The language around the use of Lemma 2.4 to "verify the results" needs to be appropriately qualified. The autograd-based method does not provide any sort of guarantee / verification, as the maximizing solution to (7) may not be found in practice. This is *at best* a sanity check for the Lipschitz bounds, but certainly not any guarantee.

[1] https://arxiv.org/abs/2510.23012

[2] https://arxiv.org/abs/2104.07167

**Audience:**

Yes

**Audience Explanation:**

This survey is thorough and likely to be of value to some of TMLR's audience. Verified robustness of neural networks, while not really scaling to deep models, also has some industry interest in the small parameter count setting.

**Broader Impact Concerns:**

I have no ethical concerns.

**Claims And Evidence:**

Yes

**Claims Explanation:**

This is challenging to evaluate because, as a survey paper, there are limited original claims. I've identified two in S1 above. I haven't checked the proofs in detail. The TMLR guidelines for survey papers write that such works should "draw new connections, highlight trends, and suggest new problems in an area." I'd say this work does a solid job of discussing connections between different applications of Lipschitzness, but doesn't do a thorough analysis of trends and promising research areas.

**Requested Changes:**

These changes are critical:
* The autograd sanity check mentioned in W5 must be correctly qualified.

These changes would strengthen the work:
* Address weakness 1, adding more context around theorem statements.
* Weaknesses 3 & 4 could be addressed, but would be significant additional effort and are not necessary to make the work valuable.
* The concurrent result [1] above should be cited
* Capitalization of section titles should be consistent (e.g. 2.3.2 vs 2.4)
* What is "dvec" in Def 2.19?

---

### Review · Reviewer_Rbf2 · 2026-02-05

**Summary Of Contributions:**

This paper provides a comprehensive review of Lipschitz Continuity and its role in modern data science (e.g., deep learning and large language models). It covers basic concepts, theoretical development, Lipschitz constant estimation, and network architecture designs to ensure Lipschitz continuity. I believe such a survey is valuable to the community.

While I appreciate this survey and enjoy the review content covered, I think there is room for improvement
1. Notation inconsistency. For example, the activation function uses notations $\sigma$, $\rho$ and $\phi$ in different sections. $\phi$ is also used for linear mapping in equation (64)
2. Repeated definition. For example, the MLP is defined in eq 63 and later again in eq 156.
3. Almost all equations are numbered, which is unnecessary
4. Section 2.6.2. Kim et al 2021's result of global Lipschitz is not about the dot product attention, but the L2 attention proposed by their paper. I believe this section should be moved to section 4.5.3
5. Unexplained notation or concepts. For example, supervision noise on page 19, v and tau in eq 231
6. The paragraph above eq 144. It misses several "$\in$" symbols.
7. Eq 166. I think this is wrong. Did the author mean  $d|W|_{op}$?
8. Eq 167 is wrong. It should be $K^{-1}$ or $K$, the relevant discussion is hence not appropriate. Also, it is better to mention WGAN (Wassertain GAN) in the paragraph above this equation.
9. After equation 171, see 2.5 -> see Section 2.5.
10. eq 188 is wrong. It should be $F^{-1}(y)$
11. eq 198. Missing a factor on the RHS?
12. paragraph above and after eq 203. Many typos in subscript and superscript
13. equation 236/237, missing a factor of Lip(g) on the RHS?

**Additional Comments:**

No

**Audience:**

Yes

**Audience Explanation:**

I never read such a summary of Lipschitz continuity. I believe it is a very helpful survey. Readers in the data science/ML community who want to know recent advances should find this survey useful

**Claims And Evidence:**

Yes

**Claims Explanation:**

All material discussed in the survey has clear citations and support from the literature.

**Requested Changes:**

Please refer to my comment above. There are quite a few typos in the paper.

---

> ### Comment · Reviewer_Rbf2 · 2026-02-12
> **Additional comment**
>
> I think I missed one comment in my previous review.
>
> I hardly believe equation (46) (and what is derived based on it) is correct. I believe the contraction inequality applies to fixed $h$ functions, but not to a family of $\mathcal H$. Even for Lipschitz family $\mathcal H$, usually a covering number-type bound is needed.

---

> > ### Author Response · Authors · 2026-02-13
> > **Thank you, we're examining the proofs and bound conditions.**
> >
> > Dear Reviewer Rbf2,
> >
> > Thank you for your careful examination and for raising concerns regarding Equation (46), where we intended to provide a bound with the contraction inequality.
> >
> > We are currently re-examining the proof and the bound, including revisiting the relevant literature.
> >
> > We will update the manuscript shortly to address your concerns.
> >
> > Thank you again for the valuable comments and efforts.
> >
> > Sincerely,
> >
> > The authors

---

> > ### Author Response · Authors · 2026-02-14
> > **Revisions regarding applying the vector contraction (Maurer, 2016, Equation 1 & Corollary 1) on eq 46.**
> >
> > Dear Reviewer Rbf2,
> >
> > Thank you for your careful examination of Equation (46).
> >
> > After re-checking our derivation and revisiting the relevant literature (**A Vector-Contraction Inequality for Rademacher Complexities, Eq. (1) and Corollary 1**), we confirm that your comment is correct: the Lipschitz constant applies to a fixed function.
> >
> > We have revised issue. The key conclusion remains unchanged: the Lipschitz constant controls the generalization bound for a fixed function h, rather than for an entire family \mathcal{H}.
> >
> > Thank you again for pointing out this issue.
> >
> > The Authors.
> >
> > [1]. A. Maurer, A Vector-Contraction Inequality for Rademacher Complexities, page 3, https://link.springer.com/chapter/10.1007/978-3-319-46379-7_1

---

### Review · Reviewer_GEmp · 2026-04-02

**Summary Of Contributions:**

This paper surveys Lipschitz continuity estimation in deep learning as a unifying lens for understanding robustness, generalization, optimization and certification. It has four parts: theoretical foundations, estimation methods, regularization approaches, and certifiable robustness, also with some corrective technical results, such as revised Lipschitz constants for common activations and bounds for more general network structures. Its main message is that Lipschitz continuity is not a niche mathematical property, but a central tool for building and analysing trustworthy neural networks.

**Audience:**

Yes

**Audience Explanation:**

Trustworthiness of neural networks is an important topic in machine learning and estimating Lipschitz continuity is one key aspect of that topic.

**Broader Impact Concerns:**

Not applicable.

**Claims And Evidence:**

Yes

**Claims Explanation:**

The submission is well supported for a survey paper, especially in its formal derivations and technical clarifications.

**Requested Changes:**

I don't have major concerns on this submission. Several suggested improvements are: sharpen the survey’s main takeaways, add an explicit comparative summary of methods and trade-offs, deepen the treatment of transformers/LLMs,

---

> ### Author Response · Authors · 2026-04-06
> **Thank you for the reviews, we're now working on the revisions.**
>
> Dear Reviewer GEmp,
>
> Thank you for the reviews and valuable suggestions regarding improving the manuscript.
>
> We are currently working on the revisions and will submit an updated version of the manuscript shortly.
>
> Best regards,
>
> The Authors

---

> ### Author Response · Authors · 2026-04-09
> **Requested revisions.**
>
> Dear Reviewer GEmp,
>
> Thank you again for your thoughtful overall feedback regarding the presentation.
>
> We have revised the manuscript by using the valuable feedback.
>
> ## Regarding
> > "sharpen the survey’s main takeaways"
>
> We agree with you that we should explicitly state the main takeaways in the manuscript. We have the revised the paragraph 3 in **Introduction**, after this sentence:
> > Beyond conducting a survey, we also critically distill the existing knowledge, present it in a more accessible manner, and provide the results or proofs missing or incorrect in the literature.
>
> we added a closing sentence for stating the main takeaways:
> > The message of this survey is that Lipschitz continuity is not just a niche mathematical property; it is a fundamental principle for building and analyzing trustworthy neural networks.
>
> ## Regarding:
> > add an explicit comparative summary of methods and trade-offs,
>
> We thank the reviewer for the suggestion. We have revised the manuscript by adding the **Remarks** subsections for section **3 Estimation Methods** and section **4 Regularization Approaches**. Please refer to **Section 3.7 Remarks** and **Section 4.6 Remarks**.
>
> ## Regarding:
> > deepen the treatment of transformers/LLMs,
>
> We thank the reviewer for this valuable suggestion. Although an earlier draft contained a deepened survey of Lipschitz continuity in transformers/LLMs, we decided not to include that material in the present manuscript. At this stage, the literature in this area remains relatively limited and suffers from soundness after we reviewed the literature. The goal of this survey is to maintain rigor and accuracy, we restricted the discussion to transformer-related results whose technical foundations we could verify with confidence in math. We believe that a more comprehensive survey of Lipschitz continuity in transformers and LLMs will become possible as the literature matures in near future. We wish to write another dedicated survey for discussing the Lipschitz continuity in LLMs in future.
>
> Thank you again.
>
> The authors.

---

### Decision · Action_Editor_Yq7T · 2026-06-30

**Recommendation:** Accept as is

**Additional Comments:**

The paper generated a fair amount of discussion with reviewers to improve the manuscript. At the end of the discussion process, all reviewers agree the paper has reached a satisfactory state and support acceptance.

**Audience:**

Yes

**Audience Explanation:**

The areas covered by the paper are of broad interest to TMLR's audience, as highlighted by the reviewers.

**Claims And Evidence:**

Yes

**Claims Explanation:**

The paper provides a systematic review of Lipschitz continuity in deep learning, unifying different research directions.

---

> ### Author Response · Authors · 2026-07-01
> **Thanks.**
>
> Dear Reviewers and AE,
>
> Thank you for reviewing our work and recommending it for acceptance.
>
> We will submit the camera-ready version shortly.
>
> Sincerely,
> The Authors

---

> ### Author Response · Authors · 2026-07-14
> **Camera-ready version has been submitted.**
>
> Dear Reviewers and Action Editor,
>
> We sincerely thank you for your valuable time, careful reviews, and constructive feedback, which have helped us improve the quality of our manuscript.
>
> To ensure the quality for this camera-ready version, we have further conducted several additional rounds of checks to ensure consistency particularly in notation, correctness of the content, and accuracy of the bibliography entries.
>
> Thank you again.
>
> The Authors